



# Comparing High-Resolution Snow Mapping Approaches in Palsa Mires: UAS LiDAR vs. Machine Learning

Alexander Störmer[1,3], Timo Kumpula[2,3], Miguel Villoslada[2,3,4], Pasi Korpelainen[2,3],
Henning Schumacher[1,3], and Benjamin Burkhard[1,3]

[1]Institute of Earth System Sciences, Physical Geography and Landscape Ecology, Leibniz University Hannover, Hannover, 30167, Germany
[2]Department of Geographical and Historical Studies, University of Eastern Finland, Joensuu, 80101, Finland
[3]Kilpisjärvi Biological Station, University of Helsinki, Kilpisjärvi, 99490, Finland
[4]Institute of Agriculture and Environmental Sciences, Estonian University of Life Sciences, Tartu, Estonia

*Correspondence to:* Alexander Störmer (stoermer@phygeo.uni-hannover.de)

**Abstract.** Snow cover has an important role in permafrost processes and dynamics, creating cooling and warming systems, impacting the aggradation and degradation of frozen soil. Despite theoretical, experimental, and remote sensing-based research,

comprehensive understanding of small-scaled snow distribution at palsas remains limited. This study compares two approaches to generate spatially continuous, small-scale snow distribution models in palsa mires in northwestern Finland based on Digital Surface Models: a machine learning approach using the Random Forest algorithm with *in-situ* measured snow depth data and an Unmanned Aerial System (UAS) equipped with a Light Detection and Ranging (LiDAR) sensor. For the first time, snow distribution was recorded over a palsa using a UAS. The aim is to review which approach is more precise overall and which

areas are not represented sufficiently accurate. In comparison to *in-situ* collected validation data, the machine learning results showed high accuracy, with a RMSE of 6.16 cm and an $R^2$ of 0.98, outperforming the LiDAR-based approach, which had an RMSE of 26.73 cm and an $R^2$ of 0.59. Random Forest models snow distribution significantly better at steep slopes and in vegetated areas. This considerable difference highlights the ability of machine learning to capture fine-scale snow distribution patterns in detail. However, our results indicate that UAS data enables the study of snow and permafrost interaction at a highly

detailed level as well.

Generally, snow accumulation zones especially at steep edges of the palsas and inside cracks are recognizable, while thin snow cover occurs at exposed areas on top of the palsas. Correspondingly, areas with thicker snow cover at the edges and inside cracks act as potential warming spots, possibly leading to heavy degradation including block erosion. In contrast, areas with thinner snow cover on the exposed crown parts can act as cooling spots. They initially stabilize the frozen core under the crown

parts, but then form steep edges and expose the frozen core, leading finally to even more block erosion and degradation.



# 1 Introduction

Snow cover plays an important role in permafrost processes and dynamics. Its physical characteristics impact the aggradation and degradation of frozen soil (Barry, 2002). In March 2023, around 39.26 million km$^2$ of the northern hemisphere were permanently or partly covered by snow (NOAA, 2023), affecting around 14.77 million km$^2$ permafrost area (Ran et al., 2022). This

includes the discontinuous permafrost areas in northern parts of Sweden, Norway and Finland, known as Fennoscandia. Due to changes in climate, a reduction in snow cover duration and an increase in snow depth have been monitored in these regions (IPCC, 2023), leading to changes of air and soil temperature interactions and resulting in negative impacts for permafrost soils (Chen et al., 2021). This has a direct impact on ecological processes, such as a reduced albedo in winter that leads to a higher energy uptake by soils (Thackeray and Fletcher, 2016) and longer growing seasons (Madani et al., 2023). In addition, Wang

et al. (2024) showed that snow cover – in combination with landscape heterogeneity - plays an important role in controlling soil temperatures throughout the year. The Intergovernmental Panel on Climate Change (IPCC) already highlighted in their latest report that a loss of permafrost within this century is expected in these regions (IPCC, 2023). Especially in northern Fennoscandia – with a main focus on northern Finnish Lapland in this study - specific periglacial permafrost landform types called palsas are in danger of disappearing within this century (Leppiniemi et al., 2023).

The occurrence of palsas, small mounds up to 4 - 7 m height with a frozen core, is bound mainly to the presence of peatlands (Meier, 2015; Seppälä, 2011). Palsas are highly sensitive to shifts in temperature insulation induced by snow during winter. Their development is directly influenced by the variety of snow depth through thermal isolation dynamics in winter and protection against warm air and sunlight in summer, leading to the growth of the Active Layer Thickness (ALT) and warming of the soils (Park et al., 2015; Verdonen et al., 2023, 2024). In detail, deep snow depth in winter leads to a lower impact on the under-

lying frozen soil due to a decreased penetration of low temperatures, while in summer this would protect the permafrost from thawing. Lower snow depth would have the exact opposite impact throughout the seasons. This influence on palsas has been empirically demonstrated by Seppälä (2011), who described the impact of variable snow cover on palsa development during subsequent thawing periods. Moreover, experiments show that mainly the depth of snow cover influences the development of palsas (Seppälä, 1982). Deviations from the usual thickness of snow cover, whether thinner or thicker, induce varying condi-

tions for palsa dynamics. For example, at the steep edges of palsas, the accumulation of snow can pose a risk by destabilizing the frozen core due to increased thermal insulation at these specific areas leading to a higher risk of block erosion (Olvmo et al., 2020; Seppälä, 1994). Despite theoretical, experimental, and remote sensing-based research, comprehensive understanding of actual snow distribution conditions within and around palsas remains limited (Seppälä, 2011; Verdonen et al., 2023). Although they have high potential for accurate mapping, Unmanned Aerial Vehicle (UAS) have not yet been used to measure snow depth

in palsa mires.

Consequently, detailed data on snow distribution in palsa mires is not available. Even if sufficient climate data is made available through official weather stations, it is the microclimate inside these mires that impacts the snow distribution in various ways, especially by snow drifts due to strong winds, as monitored exemplarily by Zuidhoff (2002). Since palsa mires occur mostly in remote areas, simple interpolation of climatic observations from weather stations within the same region does not provide





data on the actual state within these mire complexes, which does not allow to monitor the exact snow distribution (Verdonen et al., 2023). Microtopography affects snow depth and creates an environment, in which the palsas usually receive enough penetrating cold air to remain stable and to last year after year due to a thin snow cover. However, warming is predicted to increase precipitation in Fennoscandia (IPCC, 2023), leading to higher snow layers in winter, which is detrimental to the palsas and contributes to their thawing. Based on snow depth mapping, identifying cold or warm spots inside palsa mires is possible

and can help to improve our understanding of further palsa dynamics. Only *in-situ* measured data can provide clear insights into these conditions. However, to date and to our knowledge, no small-scale mapping of snow depth in palsa mires has yet been carried out.

Measuring snow depth manually demands a relatively high workload in time and effort under mostly harsh climatic conditions. Thus, measurements of snow depth over a long time span without technical help have been undertaken in Finnish Lapland by

only a few researchers, e.g. by Leppänen et al. (2016), who describe the snow survey program by the Finnish Meteorological Institute (FMI) that was established in 1909 in Sodankylä. Statistical evaluations of collected data at weather stations were published for whole Finnish Lapland (Merkouriadi et al., 2017).

The use of remote sensing data and methods to monitor snow depth has become increasingly important in snow research. Satellite data has been widely used for many years to monitor and estimate snow properties such as snow density or snow water

equivalent (Holmberg et al., 2024), but only in coarse resolution due to the technical limitations of satellites. Numerous studies have been published recently on methods to determine snow depth with Unmanned Aerial Systems (UAS) in high resolution (cell size of 10 cm x 10 cm, respectively 5 cm x 5 cm), e.g. by Bühler et al. (2016) Bühler et al. (2016) and Michele et al. (2016), who directly mapped snow depth in alpine terrain using UAS RGB. Rauhala et al. (2023) and Meriö et al. (2023) compared different UAS in terms of snow depth mapping precision for a test site in Finnish Lapland. They concluded that spatially

representative estimates of snow depth can be obtained with UAS. Furthermore, several studies utilizing satellite images to map snow distribution were conducted (Marti et al., 2016; Hu et al., 2023). However, coarse satellite datasets significantly limit the resolution of the results, especially to capture the ground surface precisely without vegetation. Small-scale structures such as palsas have very strong changes in snow depth at a fine spatial scale, which limits the information value of satellite data on small-scale processes in these structures. UAS-based data can provide data to close this gap, but these techniques also have

difficulties in mapping the ground surface without vegetation, particularly in subartic areas with low but dense vegetation.

Another method to estimate snow distribution is the application of statistical machine learning models like the *Random Forest* (RF) algorithm (Breiman, 2001), which – in snow research - has so far mainly been used for large-scale mapping of snow distribution in mountainous regions (Meloche et al., 2022; Revuelto et al., 2020; Richiardi et al., 2023). However, these studies have not been conducted at a small scale with very high resolution (< 1 m x 1 m cell size), and – to our knowledge - no study

has been conducted in subarctic and permafrost areas such as Finnish Lapland.

In this study, we applied a combination and comparison of all three mentioned methods by assessing three exemplary palsa sites in northernmost Finland: i) precise snow depth data measured in the field; ii) snow distribution calculated with UAS data; and iii) simulated snow distribution by a RF approach. The field measured snow depth data was used to validate the results given by remote sensing devices and simulation models. The aim is to identify the most promising approaches for accurately



determining small-scale snow distribution and to test a method for generating precise snow distribution maps. Specifically, this study evaluates whether machine learning can effectively model small-scale snow distribution, providing snow researchers with a cost-effective alternative to expensive UAS-based methods for future research. Furthermore, we provide a detailed assessment about the distribution of snow depth at palsas and their surrounding areas at a small scale. Therefore, this paper aims to answer following research questions: How is snow depth distributed at palsas at a small scale and is it possible to identify warming and cooling spots based on the results (1) and how accurate are snow distribution analyses based on UAS data and the RF algorithm, and which are the most suitable input parameter for the modeling approach (2)?

The obtained results provide novel insights of precise snow distribution in palsa mires including knowledge about warming and cooling spots and its negative and positive effects. Furthermore, this information can improve our understanding of the paradoxical effects of changing snow distribution due to global warming on the palsas.

## 2 Study sites

The palsa sites under investigation - Puolikkoniva, Pousu and Peera - are located ca. 30 km south from the closest Finnish Meteorological Institute's (FMI) weather station in Kilpisjärvi, Finland (Fig. 1). These sites are located along the Könkämäeno river, a significant terrain depression with numerous palsas, and are adjacent to the region's primary main road (European Route E8) in the northwestern part of Finnish Lapland (Fig. 1 (a, b)). While Peera has been previously described by Verdonen et al. (2023), Pousu and Puolikkoniva were not yet investigated.

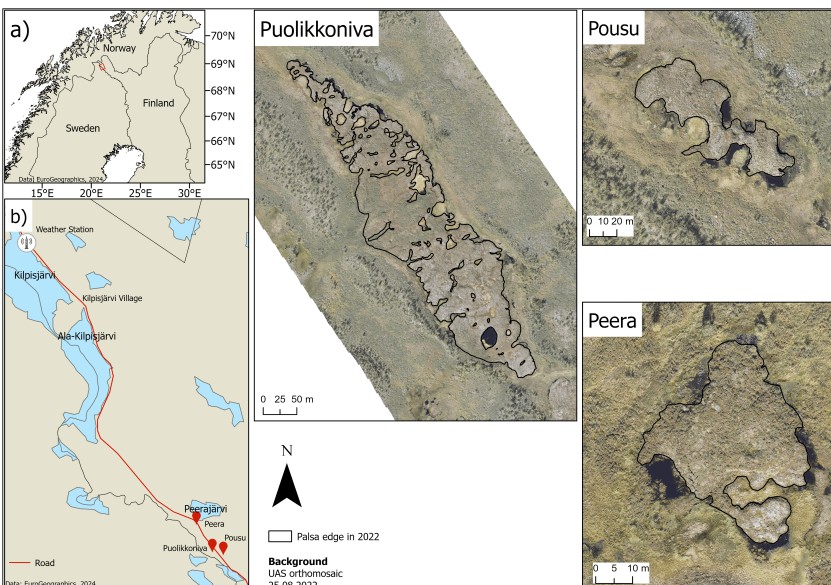

**Figure 1.** Location of the study sites Puolikkoniva, Pousu and Peera in north-western Finland (a). Climate data used in this study are from the Kilpisjärvi weather station, from which the distance to the palsa sites is around 20 km (b). Basemaps obtained from EuroGeographics (2024).



## 2.1 Palsa mire sites

**Puolikkoniva** is located approximately 2.3 km south of lake Peerajärvi at 68°51'43" N, 21°06'18" E and around 455 m a.s.l,
surrounded at the eastern part by the Peerasuvanto river and in the west by the main road. The study site has an area of roughly
4.26 ha with a maximum height difference to the surrounding peatland of ca. 2 m. This palsa is about 590 m in length and
130 m in width. Puolikkoniva is the largest study site and a prototypical longitudinal plateau palsa, consisting of several single
and complex shaped palsas. According to the definition of Seppälä (2006) and Meier (2015), the palsas contain a perennially
frozen core of peat with segregated ice. Numerous cracks traverse the palsa site, with dwarf shrubs (e.g. *Rubus chamaemorus,
Empetrum hermaphroditum*) and birches (*Betula nana*) at the edges, while atop and around the palsas, typical vegetation such
as lichens (e.g. *Cetraria spp., Cladonia spp.*) and sphagnum moss (e.g. *Sphagnum lindbergii*) flourishes. The absence of distinct
dome-shaped structures and the presence of thermokarst ponds within the palsa structure indicate the degradation of the palsa
(Seppälä, 2011), with pronounced block erosion at steep edges.

**Pousu** is located approximately 600 m east of the Puolikkoniva palsa at 68°51'39" N, 21°07'17" E and around 470 m a.s.l.,
150 m east of the main road. This study site covers an area of about 0.36 ha and has a maximum height difference to the
surrounding peatland of ca. 2.5 m. This site, measuring 130 m in length and 50 m in width, is a classic example of a degrading
dome-shaped palsa, as shown by collapsed parts, block erosion at its steep edges and thermokarst ponds within the former
palsa structure. Similar to Puolikkoniva, typical palsa mire vegetation grows around and on it.

**Peera** is located at 68°52'45" N, 21°04'35" E and around 460 m a.s.l., approximately 400 m south of lake Peerajärvi and 100
m west of the main road. This study site encompasses an area of about 0.13 ha and has a maximum height difference to the
surrounding peatland of ca. 2 m. This palsa is about 55 m in length and 45 m in width. The palsa structure is surrounded by
typical peatland vegetation such as sphagnum mosses. Water bodies, peat and bare rock structures can be found at the edges of
the palsa. Mainly lichens and dwarf shrubs grow on top of the palsa. Similar to Pousu, this palsa is also dome-shaped and in a
degrading phase. (Verdonen et al., 2023) point out a massive loss of the permafrost area during the past 15 to 60 years.

**Table 1.** Main characteristics of each palsa site. Pictures recorded with DJI mini 3 Pro UAS.

| | Puolikkoniva | Pousu | Peera |
|---|---|---|---|
| UAS picture from 30.08.2023 | | | |
| Location | 68°51'43" N, 21°06'18" E | 68°51'39" N, 21°07'17" E | 68°52'45" N, 21°04'35" E |
| Area | 4.26 ha | 0.36 ha | 0.13 ha |
| Extent (length, width) | 590 m, 130 m | 130 m, 50 m | 55 m, 45 m |
| Height | 2 m | 2.5 m | 2 m |



## 2.2 Climate

The investigation areas are located on the pre-alpine belt of the Scandes. The annual mean temperature is -1.38 °C, the annual
mean precipitation amount is about 514 mm and the dominating wind direction is south-southeast from November to April
(FMI, 2022). Higher mountains affect the local weather conditions, clouds for instance remain in front of the mountain or wind
directions are influenced (Autio and Heikkinen, 2002). This may lead to different precipitation amounts or wind directions and
speeds than measured at the Kilpisjärvi weather station (Verdonen et al., 2023). Also, high wind speeds during winter can lead
to a more intensive snow drift, influencing the snow depth distribution inside the mire sites (DeWalle and Rango, 2008).

The palsa mire sites are affected by cold winters and moderate warm summers (Fig. 2). Winter is the longest season, lasting
about 200 days including the polar night with around 50 days without sunlight. During winter the temperature can drop close
to - 50 °C and can increase above 0 °C (FMI, 2024). The duration of permanent snow cover lasts about 217 days a year (Lépy
and Pasanen, 2017). During spring, the snow cover melts away, and the growing season starts in late May. In late August the
growing season ends with the beginning of autumn which lasts around 102 days (Kauhanen, 2013).

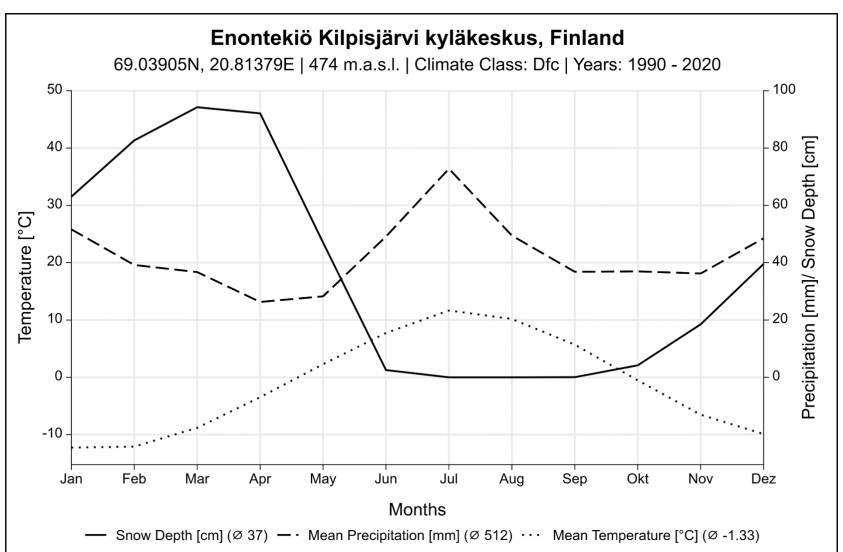

**Figure 2.** Climate chart of Kilpisjärvi (FMI, 2022). Dotted line shows 2 m above ground temperature in °C, dashed line shows precipitation
in mm and solid line shows snow depth in cm.

## 3 Data and methods

In August 2022 and March 2023, field expeditions to the palsa mire sites were conducted to collect a comprehensive dataset.
The selection of August for the summer dataset collection was strategic, corresponding with the peak of the growing season
and the maximum ALT (Verdonen et al., 2023). This timing ensures the capture of the landscape's conditions in its most diverse
states before the start of the winter season, providing the basis for extracting relevant input parameters for our approach.



On the contrary, the winter dataset, collected in March 2023, was chosen based on historical climatological patterns in the Kilpisjärvi region, which typically have maximum snow depths at this time (FMI, 2024). This period allows the collection of data under conditions that reflect winter extremes, which serves as both validation and training data for the RF modeling. Figure 3 shows an overview of the different steps carried out for this work.

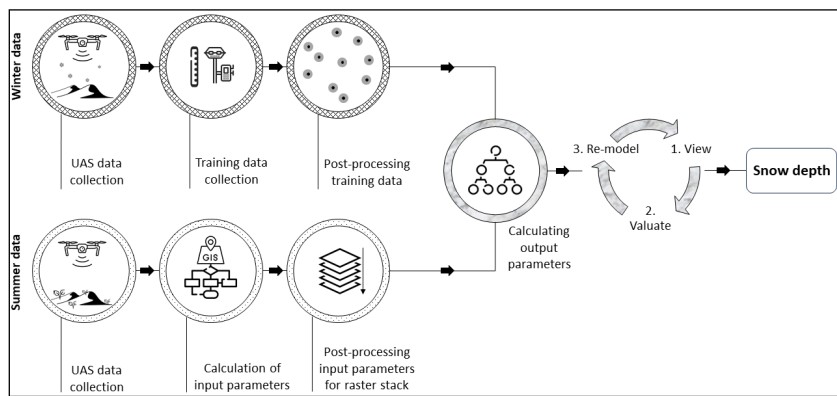

**Figure 3.** General overview of the data collection and analysis.

## 3.1    Data collection

For the collection of input parameter data, aerial surveys were conducted at all three study sites during summer using a DJI Matrice 300 RTK, equipped with a YellowScan Mapper+ LiDAR system that scanned at a wavelength of 905 nm. To improve the accuracy of the collected data, Ground Control Points (GCPs) were set using a Trimble R12i Real-Time Kinematic (RTK) GNSS. In addition, with the integrated high-resolution RGB sensor of the DJI Matrice 300 RTK, orthopictures were captured, enabling a comprehensive analysis of the sites' conditions.

These aerial missions were post-processed in detail using YellowScan CloudStation Agisoft Metashape Professional Software, resulting in Digital Surface Models (DSMs) with a raster cell resolution of 0.1 m x 0.1 m. No additional filtering or noise removal was performed on the UAS LiDAR data, keeping vegetation in the DSMs. Structure from Motion techniques were not applied and no imagery was used in the post-processing. Orthophotos were also created to provide detailed visual representations of the terrain. Mean point cloud densities per raster cell vary from 7.2 (summer) to 9.1 (winter) for Peera site, 9.9 to 8.1

four Pousu site and 6.6 to 9.0 for Puolikkoniva site.

   The winter survey replicated the methodological framework of the summer survey, using the same UAS and sensor configurations to produce DSMs and orthophotos. Based on the summer and winter DSM of the palsa sites, snow distribution datasets were calculated by substracting the winter by the summer DSM in Geographic Information Systems (GIS) - *ArcGIS Pro* by Esri was used -, allowing the comparison of UAS-LiDAR conducted snow cover ($SD_{LiDAR}$) and RF modeled ($SD_{RF}$). Additional

datasets that are essential for modeling and validation were collected after the respective flights. Snow depth measurements ($SD_{in-situ}$) were carried out manually using a yardstick across all sites, whereas each point was measured by RTK. A total of



validation points were recorded, divided across the sites as follows: 100 in Puolikkoniva, 46 in Pousu, and 39 in Peera (Fig. 4).

To ensure the derivation of an optimal $SD_{in-situ}$ training dataset, different measurement network designs were attempted at each
170 site, customised to the unique geomorphological features of the palsa mires, making sure to catch points on top of the palsa, at the edges and at the steep slopes, on the thermokarst ponds and the surrounding field as at those parts differences in snow depth can be expected. In Pousu, a randomised sampling strategy was applied with focus on the palsa edges and summits (Fig. 4 b). In Puolikkoniva, there were two parallel transects measured following its longitude shape and complemented by randomised points at the edge, thermokarst and surrounding field (Fig. 4 a). The Peera approach consisted of two intersecting
transects, augmented by a set of randomly chosen points along the edge (Fig. 4 c). This training dataset captures the variability of snow cover within palsa mires, ranging from snow-free palsa summits to deeply covered palsa edges. A histogram of the snow measurements can be viewed in Appendix A1.

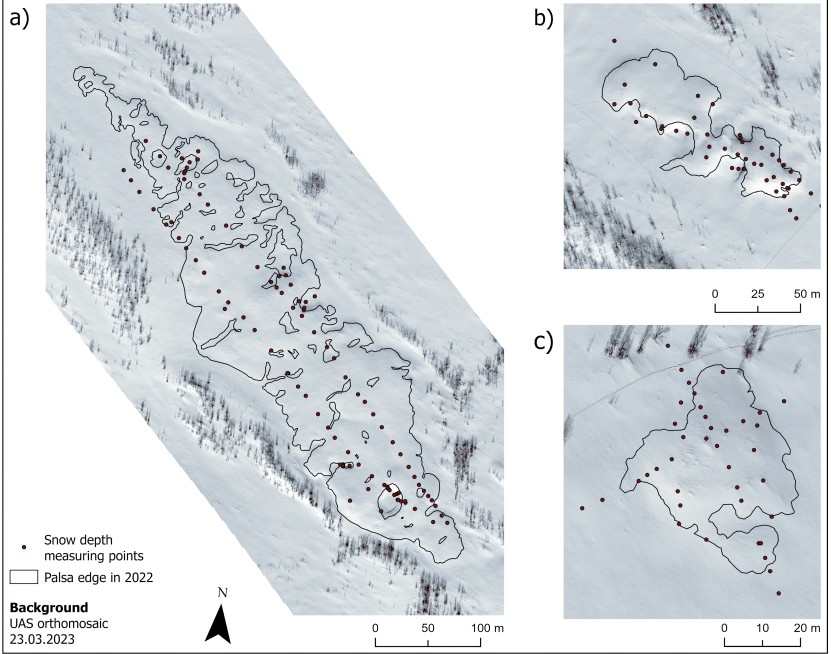

**Figure 4.** Snow depth measuring points within the investigation sites at Puolikkoniva (a), Pousu (b) and Peera (c) palsa illustrating different methods for recording snow depth (transects, randomized, crossed).

## 3.2 Random Forest algorithm

For the modeling process, we used the **ranger** package (Wright, 2023) within the R programming environment, which is known
for its ability to efficiently process large datasets and accounts for complicated predictor interactions. The preparatory steps included aggregating the various input parameters into a uniform raster stack, conducted by the *stack* function from the **raster**



package (Hijmans, 2023). This method allowed for a comprehensive analysis of the multidimensional dataset, ensuring spatial alignment across all layers.

The dependent variable for our model was the previous described $SD_{in\text{-}situ}$ dataset. These measurements, together with the stacked input parameters as independent variables, formed the basis of our RF model. The process of extracting input parameter values from the stacked raster set was performed using the buffered shapefile. This procedure was crucial for the preparation of the training dataset, which was subsequently split randomly into training (70%) and testing (30%) subsets for all palsa locations combined. This split allowed a precise evaluation of the model's predictive accuracy.

The RF model was used without explicit hyperparameter specifications, so default settings were used, including the construction of 500 decision trees, a maximum of 3 variables per split, and a target node size of 5, reflecting the characteristics of the final model run (12 input parameters). Permutation mode was chosen for variable importance assessment, and a specific seed value was implemented to ensure reproducibility of the results. The resulting Permutation Importance (PI) value of each input parameter was given and - for better understanding and comparability - the PI values were converted into percentages over all input parameters. Subsequently, the trained RF model was employed to calculate $SD_{RF}$ predictions across each palsa site. For that, the *predict* function was used, applying the model to the test dataset to estimate snow depth values.

### 3.3 Modeling data preparation

The collected airborne data were subjected to extensive preprocessing using GIS, in particular *ArcGIS Pro* and *SAGA GIS* by SourceForge. This preprocessing aimed to match the datasets (cell size 0.1 m x 0.1 m), making it suitable for analysis with the RF algorithm.

Subsequently, *SAGA GIS* was used for the computation of various geomorphological parameters to enhance the training dataset with a diverse range of topographical and environmental predictors. These predictors included elevation, aspect, slope, and a range of indices comprising hydrological and morphological landscape features (Table 2). According to Meloche et al. (2022) and Revuelto et al. (2020), the *Topographic Position Index* (TPI) is most suitable due to its proven relevance to show dependencies between topography and snow depth.

To reduce the risk of possible overfitting, the *Elevation* parameter was purposely excluded in the subsequent model iteration. Further refinement was based on the RF algorithm's PI values, leading to the exclusion of parameters with initial minimal impact on model performance (PI < 1.25 %) for the final model iteration. These parameters included: *Analytical Hillshading, Convergence Index, LS Factor, Plan Curvature, Profile Curvature, Real Surface Area, Terrain Ruggedness Index, Topographic Wetness Index*, and *Total Catchment Area* (see Table 2).

The $SD_{in\text{-}situ}$ locations, serving as the training dataset, were buffered by 0.3 m, with each buffered point assigned the corresponding snow depth value. This buffering strategy aimed to moderate model variability and offers a balanced representation of parameter combinations linked to the specific $SD_{in\text{-}situ}$ measurements, improving the realism and consistency of the model as proven in Bergamo et al. (2023).



**Table 2.** Overview of all input parameter used in the RF modeling.

| Parameter | Description |
|---|---|
| *Analytical Hillshading* | Parameter shows landscape as shaded by the sunlight from a specific direction (Tarini et al., 2006). Inserts topographical structures to the training data set. |
| *Aspect* | Aspect in degree of every raster cell (Olaya, 2009). |
| *Channel Network Base Level* | Gives information channel networks and interpolates the base level elevations of it (Olaya and Conrad, 2009). |
| *Channel Network Distance* | Gives information about the vertical distance from altitudes above the channel network to its base (Olaya and Conrad, 2009). |
| *Convergence Index* | Shows an index of convergence in relation to the overland flow (Kiss, 2004). |
| *Elevation* | The elevation calculated from remote sensing data in Agisoft Metashape Professional. |
| *LS Factor* | The factorised length and steepness of a slope (Böhner and Selige, 2006). |
| *Negative Openness* | Parameter which indicates how enclosed the location of a landscape is (Yokoyama et al., 2002). |
| *Plan Curvature* | Describes the curvature of the surface, with positive values indicating areas of convergent and negative values divergent flow (Wood, 1996). |
| *Positive Openness* | Parameter which indicates how dominant the location of a landscape is (Yokoyama et al., 2002). |
| *Profile Curvature* | Describes the curvature where the Z axis intersects with the direction of maximum gradient, with positive values indicating convex and negative values concave profile (Wood, 1996). |
| *Real Surface Area* | Shows real cell area and considers texture of the surface (Grohmann et al., 2009). |
| *Relative Slope Position* | Providing a measure of each cell's position in relation to the surrounding terrain (Böhner and Selige, 2006). |
| *Slope* | Slope in degree of every raster cell (Olaya and Conrad, 2009). |
| *Terrain Ruggedness Index* | Quantifies the terrain ruggedness with higher values representing higher roughness (Riley et al., 1999). |
| *Topographic Position Index* | Parameter based on Guisan et al. (1999), which combines several topographic features (Wilson and Gallant, 2000). |
| *Topographic Wetness Index* | Parameter which maps the relative wetness or moisture potential of each cell with higher values indicating wetter areas (Böhner and Selige, 2006). |
| *Total Catchment Area* | Represents the overall area draining towards a specific point, reflecting the vulnerability of an area to collect water respectively snow (Gruber and Peckham, 2009). |





| Valley Depth | Calculates the depth of valleys by finding the difference between each cell's elevation and an interpolated ridge level, where positive values indicate areas below the interpolated ridges, representing valleys, and negative values describing elevated regions like hills or ridges (Conrad et al., 2015). |
|---|---|
| Wind Effect | Index indicating values below 1 for shaded areas and above 1 for exposed areas in relation to a specific wind direction (Gerlitz et al., 2015). |
| Wind Exposition | Like parameter *Wind Effect*, but it automatically calculates an Index taking all wind directions into account (Gerlitz et al., 2015). |

## 3.4 Statistical analysis

The statistical analysis focused on evaluating the predictive accuracy of the model. The following metrics were summarized for all palsa sites and calculated in the R environment:

1. **Coefficient of Determination ($R^2$):** Calculated to quantify the proportion of variance in the dependent variable that is predictable from the independent variables in the model (Nagelkerke, 1991), giving clearance about the overall effectiveness of the model, defined as

$$R^2 = 1 - \frac{\sum (y_i - \hat{y}_i)^2}{\sum (y_i - \bar{y})^2} \tag{1}$$

2. **Root Mean Square Error (RMSE):** Employed to quantify the average magnitude of the error in the predictions (Chai and Draxler, 2014), highlighting the ability of the model to predict snow depth accurately, defined as

$$\text{RMSE} = \sqrt{\frac{1}{n} \sum (y_i - \hat{y}_i)^2} \tag{2}$$

3. **Mean Absolute Error (MAE):** Measures the average magnitude of the absolute errors between predicted and observed
values, without considering their direction (Chai and Draxler, 2014; Willmott and Matsuura, 2005), defined as

$$\text{MAE} = \frac{1}{n} \sum_{i=1}^{n} |y_i - \hat{y}_i| \tag{3}$$

4. **Standard Deviation (SD):** Provides a measure of the dispersion of prediction errors around their mean (Walser, 2011), revealing the precision and consistency of the predictions, defined as

$$\text{SD} = \sqrt{\frac{1}{n-1} \sum_{i=1}^{n} (y_i - \hat{y}_i)^2} \tag{4}$$

where $y_i$ is the observed value, $\hat{y}_i$ is the predicted value from the model, $\bar{y}$ is the mean of observed values.





Furthermore a 10-fold cross-validation according to James et al. (2013) was done in RStudio using the **caret** package (Kuhn, 2023) to further validate the model and mitigate the risk of overfitting. This method divides the $SD_{in-situ}$ training data into ten subsets, trains the model on 9 subsets, and evaluates it on the remaining one. This cycle was repeated ten times, with each subset serving as the test set once, ensuring that every data point is used for both training and testing. The choice of 10 folds balances the need for model evaluation with computational efficiency. In the end, the mean $R^2$ and RMSE were calculated, allowing the comparison with the initially calculated values.

Additionally, a correlation analysis between the input parameters and predicted $SD_{RF}$ was performed to identify any significant predictors and assess their influence on the model's predictions. Special focus was set to parameters exceeding or fall below a correlation of +/- 0.7, indicating a significant influence on the model performance.

To visualize the statistical analysis results, scatter plots of RF and UAS LiDAR derived snow depths in comparison to the measured values were conducted.

## 4  Results

### 4.1  Snow depth predictions

The predicted $SD_{RF}$ present a good visual alignment with the calculated $SD_{LiDAR}$ (Fig. 5).

The **Puolikkoniva** palsa site is affected by several collapsed areas, in which snow accumulates massively. This can be seen in the $SD_{RF}$ (Fig. 5 a) as well as in the $SD_{LiDAR}$ (Fig. 5 b) results. In general, the RF models the snow depth inside these collapsed holes and cracks slightly higher than the UAS LiDAR was detecting it. Especially directly at the steep edges of the palsa, the depth values increase up to 30 cm. However, the transition of the snow depths better corresponds to changes at slopes on the UAS LiDAR results, as the RF model reveals obvious patterns. The most obvious differences are occurring in areas beneath the palsa itself, for example the whole northeastern and eastern parts, which have higher snow depths predicted by RF than detected based on the UAS LiDAR data. The most similar parts are the areas on top of the palsa with slightly higher snow depths predicted by RF.

The **Pousu** palsa site shows a similar pattern as in the Puolikkoniva palsa site, with cracks filled by snow and collapsed parts with steep slopes where snow accumulated heavily (Fig. 5 c, d). Again, the transition of the snow depth at those areas is more natural in the $SD_{LiDAR}$ data since the $SD_{RF}$ data are showing sharp steps. Also, mire areas parts next to the palsa are observed by UAS LiDAR with lower snow depth as modeled with RF. This is especially visible in the southwestern and southern parts of the area, where snow height was modeled between 40 – 50 cm and the UAS LiDAR detected values between 10 – 20 cm. However, similarities are visible on top of the palsa, where snow depths were modeled and observed in a range between 10 to 30 cm each.

The **Peera** palsa site shows the highest consistency between $SD_{RF}$ and $SD_{LiDAR}$. However, as in the two former sites, the highest snow pack accumulated in cracks and at the steep edges of the palsa (Fig. 5 e, f). Unlike at the other locations, there are no sharp steps at the parts mentioned here, as both approaches model smooth transitions. Similar structures are also visible on top





of the palsa with snow heights around 20 cm in each approach. However, differences are visible like at the two other palsa sites

in the surrounding area of the palsa, where the snow pack is calculated higher by RF than the UAS LiDAR detected it.

**Figure 5.** Snow depth predictions based on the RF model run 3 (left) and the UAS LiDAR (right) at site Puolikkoniva (a, b), Pousu (c, d) and Peera (e, f) palsas. Red points are showing the *in-situ* snow depth measurement locations.



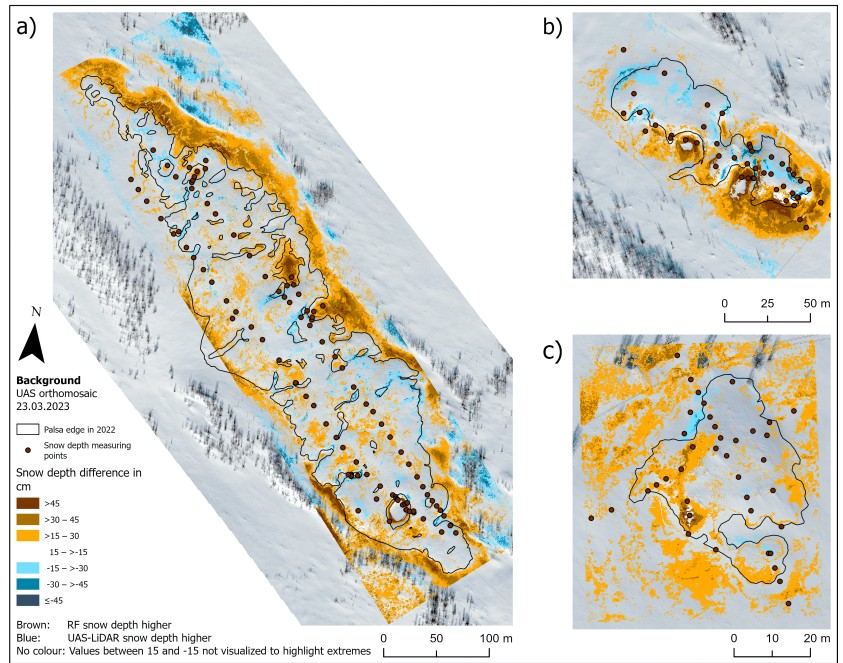

**Figure 6.** Snow depth differences between modeled and UAS LiDAR results at a) Puolikkoniva, b) Pousu and c) Peera palsas.

Viewing the deviations in snow height between the two approaches, it is evident that the top parts of the palsa sites themselves show very low differences (Fig. 6 a, b, c). However, deviations occurred at the edges, inside of cracks, at the highest parts of the palsa and in the surrounding areas. Within cracks on top of the palsa sites, the UAS LiDAR detected in general higher snow pack than RF modeled it. Differences of around 20 cm are shown, but with peaks up to 50 cm. Also, highest elevated

structures of the palsa sites directly at edges show deviations of about 15 - 30 cm higher snow pack calculated by the UAS LiDAR. In contrast, the collapsed parts with accumulated snow are consistently modeled with higher values exceeding 45 cm of deviation to the UAS LiDAR derived values. It is worth noting that the deviations in the areas surrounding the palsas are mostly in the type of higher modeled snow from the RF model. Only a few narrow structures with significantly higher snow can be recognized based on the UAS LiDAR data.

## 275 4.2 Variable importances

The calculated PI shares of all parameters for all model runs are pictured in Fig. 7. In model run 1, the four most important parameters are *TPI, Wind Effect, Valley Depth and Channel Network Base Level*, while *TPI* is nearly three times more important (29.36 %) than the second most important parameter with 9.33 %. Low importances are shown for the parameters *Terrain Ruggedness Index, Real Surface Area, Analytical Hillshading, Profile Curvature, Topographic Wetness Index, LS Factor, Total*

280 *Catchment Area, Plan Curvature* and *Convergence Index*. These parameters showed a PI value lower than 1.25 %, which is about 24 times less than the most important parameter *TPI*.



After removing the parameter *Elevation* for model run 2 and all parameters with PI values lower than 1.25 % for model run 3, slight changes in PI values can be recognized. The remove of *Elevation* led to a minor PI value increase for 15 parameters and slight decrease for 5 parameters. However, removing all parameters with PI values lower than 1.25 % increased the importance of all remaining parameters, especially for the four most important parameters and *Elevation* as well as *Wind Exposition* with 8 - 15 % total increase.

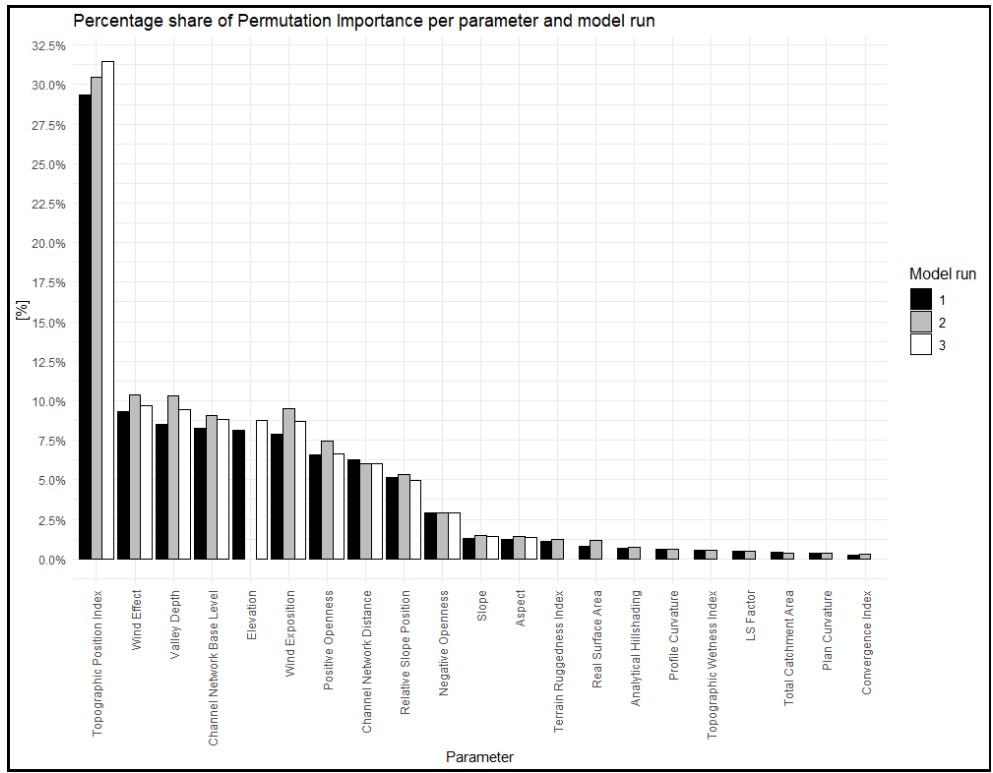

**Figure 7.** Overview of percentage shares of PI per input parameter and model run.

## 4.3 Statistical evaluation results

The statistical analysis by model run (Table 3) and validation point locations (Table 4) reveals high precision of $SD_{RF}$. While each model run has significantly better metrics than the $SD_{LiDAR}$, only minor differences are found between model runs. The RMSE amounts to 6.16 cm for model run 3, which is about 1.2 cm better than the worst model run 2. Same tendencies can be seen for $R^2$, MAE and SD with model run 3 having the best and model run 2 the worst results among all. However, the $SD_{LiDAR}$ results drop off in accuracy for each metric compared to the model runs with 20 cm worse RMSE, a $R^2$ of 0.59 and MAE respectively SD with ca. 15 cm and 19 cm higher values.

Separated to point groups, the results of $SD_{RF}$ show clear trends among the individual groups. While RMSE, MAE and SD is in group *Open Area* best (1.42 cm, 0.89 cm, 1.39 cm), followed by *Thermokarst* (1.44 cm, 0.98 cm, 1.42 cm), *On Top* (1.79



cm, 0.99 cm, 1.73 cm) and *Edge* (2.56 cm, 1.44 cm, 2.55 cm), for metric $R^2$ there are no significant differences among each group (all 0.99). However, the same metrics - but for $SD_{LiDAR}$ - were revealing significant differences. Each accuracy metric shows lower results compared to the RF performance, but also the order of point groups shows different results. Thus, group *On Top* contains the best RMSE and MAE (17.01 cm, 12.20 cm) among the groups followed by *Open Area* (17.33 cm, 14.18 cm), *Edge* (32.53 cm, 22.66 cm) and *Thermokarst* (40.73 cm, 34.04 cm), but for SD *Open Area* is best (14.20 cm) followed by *On Top* (16.87 cm), *Thermokarst* (24.21 cm) and *Edge* (30.46 cm). The $R^2$ metric shows a completely different order with best results for *Edge* (0.48) followed by *Thermokarst* (0.22), *Open Area* (0.16) and *On Top* (0.00).

**Table 3.** Overview of calculated Root Mean Square Error (RMSE), Coefficient of Determination ($R^2$), Mean Absolute Error (MAE) and Standard Deviation (SD) for model runs 1 to 3 and UAS LiDAR.

| Parameter | Model Run 1 | Model Run 2 | Model Run 3 | UAS LiDAR Snow Depth |
|---|---|---|---|---|
| RMSE | 6.98 | 7.34 | 6.16 | 26.73 |
| $R^2$ | 0.97 | 0.97 | 0.98 | 0.59 |
| MAE | 4.39 | 4.59 | 3.25 | 18.68 |
| SD | 6.97 | 7.34 | 6.16 | 25.15 |

**Table 4.** Overview of RMSE, $R^2$, MAE and SD divided by validation point locations within the investigation areas.

| | RMSE | | $R^2$ | | MAE | | SD | |
|---|---|---|---|---|---|---|---|---|
| | *RF* | *LiDAR* | *RF* | *LiDAR* | *RF* | *LiDAR* | *RF* | *LiDAR* |
| On Top | 1.79 | 17.01 | 0.995 | 0.009 | 0.99 | 12.20 | 1.73 | 16.87 |
| Edge | 2.56 | 32.53 | 0.996 | 0.482 | 1.44 | 22.66 | 2.55 | 30.46 |
| Thermokarst | 1.44 | 40.73 | 0.999 | 0.223 | 0.98 | 34.04 | 1.42 | 24.21 |
| Open Area | 1.42 | 17.33 | 0.993 | 0.163 | 0.89 | 14.18 | 1.39 | 14.20 |

The presented scatter plots of both approaches (Fig. 8) are revealing further insights into the accuracy of the results. On the left, the plot depicts the relationship between $SD_{in\text{-}situ}$ and $SD_{LiDAR}$ and on the right, $SD_{RF}$ results are compared to it.

The UAS LiDAR approach shows a positive linear relationship between detected and measured snow depths. However, variability is visible especially at higher snow depths. The RF approach shows a strong positive linear relationship with data points closely following the trend line and exhibiting minimal deviation, meaning a higher correlation between $SD_{RF}$ and $SD_{in\text{-}situ}$ than for $SD_{LiDAR}$ can be observed.



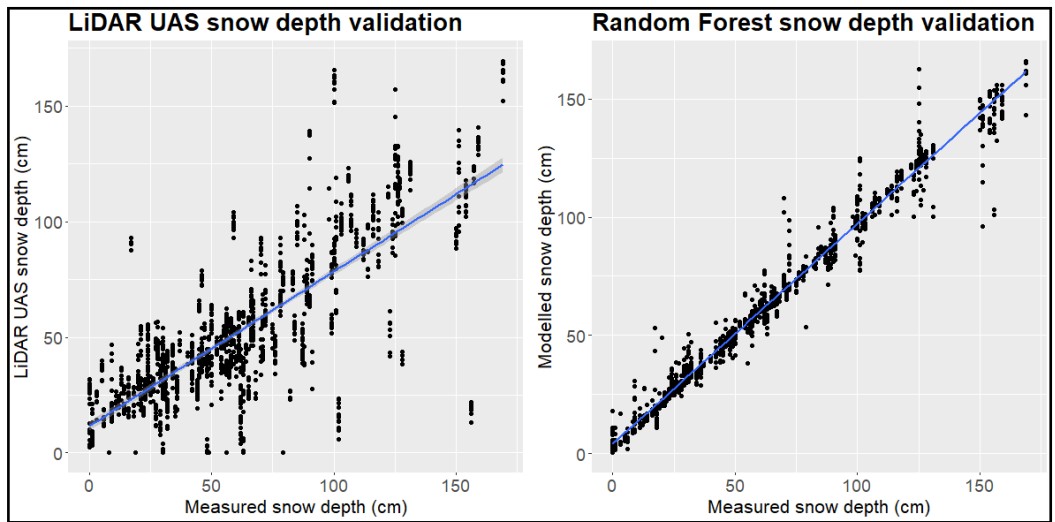

**Figure 8.** Scatter plots with regression lines of the UAS LiDAR derived and RF modeled snow depths.

The 10-fold cross-validation supports the validation results shown above and, with an $R^2$ of 0.97 and an RMSE of 6.4 cm, shows

good consistency with the statistical metrics mentioned above. In addition, the correlation analysis of the input parameters and

$SD_{RF}$ shows a high correlation for the *TPI* (-0.79), which indicates a large influence of this parameter on the model performance. However, all other parameters do not show a comparably high correlation. The parameters *Channel Network Distance* (-0.23) and *Relative Slope Position* (-0.34) are worth mentioning.

## 5   Discussion

### 5.1   Snow distribution mapping in palsa mires and its impacts

The snow cover maps provide a high-resolution and precise overview of the distribution within palsa mires. Unique small-scale differences in the height of the snowpack, which are closely related to the topographic properties of the palsa mires, are depicted in $SD_{RF}$ results. However, also $SD_{LiDAR}$ maps provide this kind of differences, although the model results are statistically more accurate.

Snow accumulation areas at palsa edges and in open cracks on the palsa tops probably have a warming impact on palsas. This effect is consistent with findings by Peng et al. (2024), who showed that snow accumulation can insulate the ground and reduce cold air penetration. Higher insulation from a dense snowpack reduces the penetration of cold air into the ground during winter. For Pousu palsa (Fig. 8 d, e), this effect can be seen at the edges in southwestern and northeastern direction, possibly because of windward and leeward effects. This leads to transportation of snow to these parts following the main wind direction which

has been south, southwest during the past 20 years in the Kilpisjärvi area (based on FMI data at Kilpisjärvi weather station). At the bottom of these edges, snow accumulates due to wind and gravitational slide down, leading to a thinner ALT at the edges




as the exposition to solar radiation is shorter due to the longer-lasting snow cover. These observations concur with findings by Verdonen et al. (2023) and Seppälä (2011). However, at accumulation zones the frost is not able to penetrate the ground deeply, meaning that the underlying soil is not able to freeze deeply. These parts of a palsa can be classified as warming spots. During late spring and summer, these areas remain more humid due to the longer snowmelt, allowing higher temperatures to penetrate deeper because of the higher thermal conductivity and destabilize the ice core especially at edges. Combined with gravitational forces due to the high slope at the edges, block erosion occurs, exposing the frozen core in late spring and summer, which then leads to increased thawing and degradation. These effects are intensified, when cracks open at the upper edge parts and are filled with snow during winter. They can be considered as warming spots as well, which lead to destabilization at the palsa edges. Martin et al. (2021) experimentally showed this in a model approach, pointing out the phase of initial slope adjustment for palsas which constantly experience snow depth between 20 – 30 cm. According to our findings, even larger depths occur at the palsa edges in the Kilpisjärvi region with increasing degradation. This cycle is repeating until the slopes at the edges are not steep enough to accumulate snow, which happens when the top plateau of a palsa is degrading as well. Snow conditions might have a more significant impact on palsa developments than previously known, as already suggested by Seppälä (2011). Further monitoring and implementation in modeling of the findings of this project can help to better understand the future palsa development.

On the other hand, cooling spots could also be identified based on the snow distribution. The top parts of the investigated palsas were covered with thin snow layers, allowing frost to penetrate the ground deeply and stabilize the ice core during winter. However, Seppälä (2003) proved that thicker snow cover on palsas prevents melting at the ice core due to a longer duration of snow cover. This means that cooling spots inhibit a greater ALT during summer, where the risk of crack occurrence is higher. As shown for the Pousu palsa (Fig. 8 d, e), the cooling spots are located near to the uppermost parts of steep edges, where the surface is heavily exposed to wind. The assumption is that this, in combination with destabilized and collapsing edges, could lead to very sharp and even vertical edges. The next stage would be the occurrence of cracks, which would lead to more block erosion and degrading of the palsa edge.





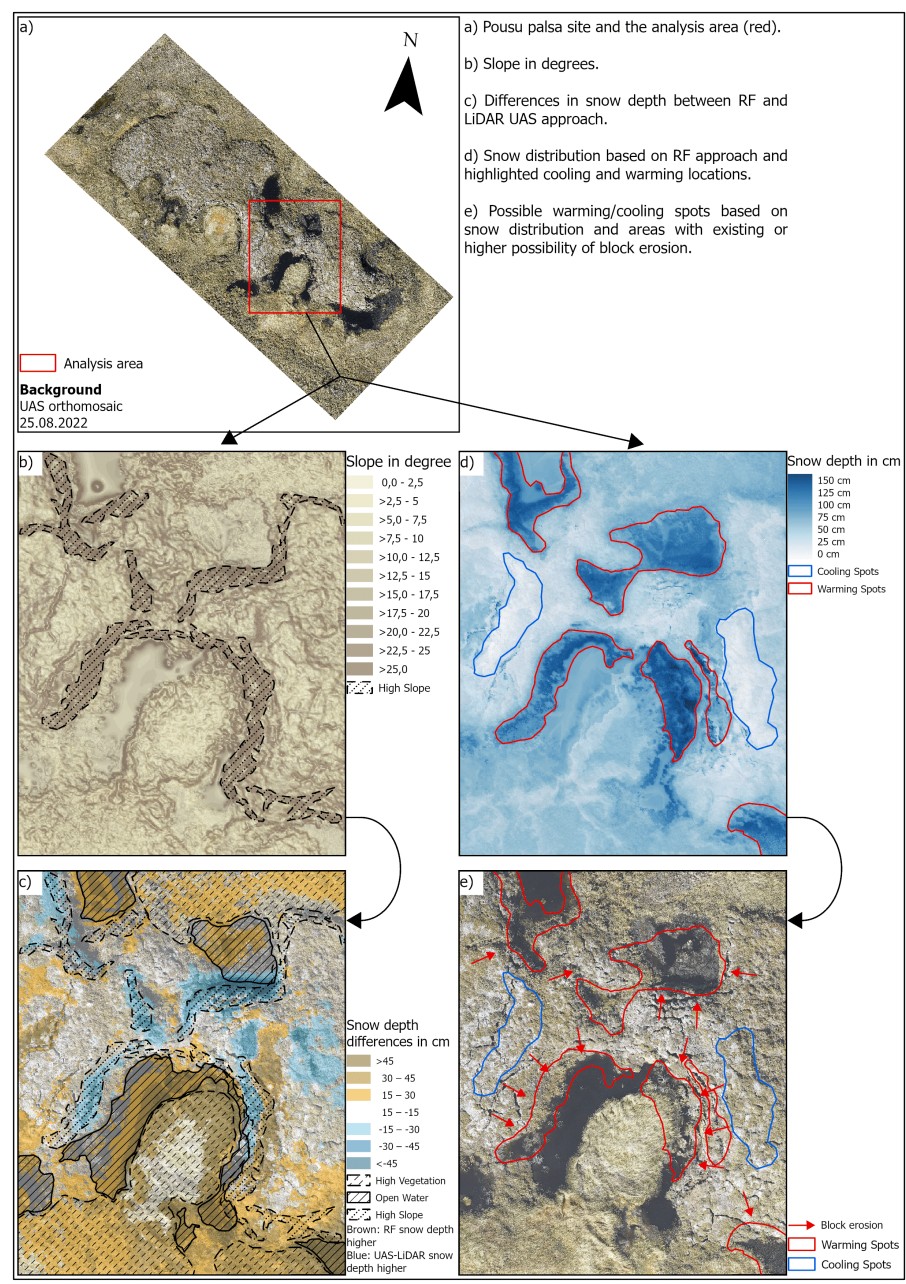

**Figure 9.** Explanation of differences between UAS LiDAR-calculated and RF-modeled snow depths.

## 5.2 Analysis of RF and LiDAR snow mapping

The statistical analysis of $SD_{RF}$ and $SD_{LiDAR}$ shows that the RF approach outperformed the latter in terms of accuracy. An overall RMSE of 6.16 cm compared to 26.73 cm reveals the better performance of machine learning algorithms compared to



the use of not in depth post-processed remote sensing data in snow distribution mapping, shown in Table 3 and Fig. 8. These findings concur with recent studies by Luo et al. (2022) and Panda et al. (2022). However, a closer look at the differences

provides further insights and helps to understand the results. Figure 5 (b, c) shows, for Pousu palsa, which kind of deviations exist between the two approaches.

It is evident that in both approaches, the upper parts of the palsa, which are mostly flat and populated by low vegetation, contain the same snow cover. High deviations occur in parts characterized by a high slope, higher vegetation, and especially open water. In summer, the vegetation within palsa mires is seasonally taller than during winter due to general growth. In

contrast, vegetation like sedges and grasses is also flattened in winter because of the snow's weight. The LiDAR sensor detects all surface elements, including the vegetation with leaves in summer, but during winter only the snow surface is recorded, leading to a bias between the summer and winter DSMs. This existing bias is also detectable in the final snow depth, which means that the snow depth is consequently calculated too low in areas with height-changing vegetation. Tall shrubs such as *Betula nana* sometimes form thickets at the palsa edges, which could cause problems to detect palsa surfaces with LiDAR.

The described problems with precise mapping of snow distribution are also described by Broxton et al. (2019). In contrast, the RF approach incorporates the existing vegetation, using the LiDAR UAS data from summer as a base for all input parameters. When combining $SD_{\text{in-situ}}$ in winter, the RF approach was able to connect between vegetation and higher snow, showcasing the model's strength in calculating these without a high bias.

The RF and UAS LiDAR approaches also show high deviations between open water areas – known specifically as thermokarst

ponds in thawing palsa mires -, although the RF models the snow depth at these parts more precisely than the latter. The low performance of the LiDAR sensor can be explained by commonly known problems with low reflecting surfaces such as water (Mandlburger and Jutzi, 2019) or the ability of detecting scattering such as snow (Deems et al., 2013). RF considers the context of all parameters in relation to the specific snow depth measured on thermokarst ponds, which explains the strong performance of the model even with faulty initial values of the UAS LiDAR data set.

Moreover, high deviations at the steep edges of the palsas are visible, here with larger depths calculated by the LiDAR sensor. This can be explained by natural degrading processes at these edges leading to differences between the summer and winter DSMs. For instance, in summer, the edges of the palsas are recorded before block erosion can partly occur and soil can slide to the bottom. In comparison with winter recordings, when block erosion occurs, the LiDAR can detect a high deviation between the DSMs. This results in unnaturally high $SD_{\text{LiDAR}}$, although the actual snow depth is lower. The same applies to cracks that

can open after the first data collection in summer, leading to higher snow depths in combination with winter data. However, these degrading processes were also not considered in the RF approach since the summer dataset was used for calculating all input parameters, indicating that the RF approach also models the snow depth at edges less accurately (see Table 4 with the most erroneous RMSE at edge parts for the RF approach). Moreover, the down-floating and accumulating of snow at edges is a chaotic process that is extremely difficult to monitor accurately. Accordingly, a natural shift in ground elevation of several

centimetres in palsa mires occurs between summer and winter due to frost heave and subsidence, as Renette et al. (2024) recently described. This should be taken into account when considering snow distribution based on two DSMs from warm and cold seasons.



These findings confirm that even with low-cost equipment such as a yardstick and moderate computer power, extremely accurate snow distribution can be modeled on a small-scale. We are aware, that in this research an expensive LiDAR sensor was used for preparing input parameter for the RF model. Therefore, it should be investigated whether a low-cost UAS RGB can provide comparable high-quality input parameters for the model or if expensive LiDAR sensors are necessary. In depth post-processing of the LiDAR dataset, like removing vegetation from the initial point cloud, can help to improve the accuracy of the conducted $SD_{LiDAR}$ to clarify the necessity of an expensive LiDAR sensor. Additionally, it is worth to find out whether a low-cost and low-quality UAS RGB can achieve the same statistical metrics for snow distribution as the UAS LiDAR in order to receive a full understanding about the necessary quality of UAS sensors for snow mapping. For large areas, or if an extremely small-scale, high-precision overview of snow depth is not needed, UAS LiDAR or RGB should be the preferred option due to the high workload and time-consuming nature of measuring snow depth. Recently, the potential of UAS imagery for snow depth estimation was explored in several studies (Marti et al., 2016; Rauhala et al., 2023; Revuelto et al., 2021).

### 5.3 Uncertainties and limitations

Snow distribution underlies erratic processes, especially because of wind drifts. This must be considered when using these methods. In addition, these machine learning models are based on a unique observation in time. Changing weather conditions could lead to a completely different snow distribution on another day, which could influence the model quality.

However, even during the initial collection of the dataset in summer and winter, errors could have occurred. Especially the LiDAR sensor is prone to errors; surface conditions with high reflectance can lead to dispersion of the laser beam and therefore to bias in the data. Moreover, not all surface elements can be fully detected due to shielding through shrub vegetation (Gould et al., 2013). This bias must be considered as a consequential error for the whole modeling approach when calculating input parameters. On the other hand, removing of vegetation in the initial LiDAR-based point cloud could improve the accuracy of $SD_{LiDAR}$, although it would increase the preparation time clearly. Also the use of different - or more - wavelengths can increase the accuracy of snow mapping with LiDAR sensors (Deems et al., 2013). Furthermore, the collection of training data also contains possible errors, although significantly lower. The yardstick is the most reliable method to measure snow depth, although dense ice layers or near ground vegetation like roots can alter the values by a few centimetres. The modeling step in RF naturally contains a high susceptibility to errors, such as overfitting. Even if the 10-fold cross-validation detects no significant overfitting, the modeling results can only be exactly verified by using measured snow depth. This should be tested in a first step before applying this method in further projects. Additionally, we have discussed the suitability of all input parameters. It is very plausible that the *TPI* has such an importance for the model because it summarizes several topographic information in one parameter. Snow accumulates at edges and drifts down slopes, meaning in terms of modeling that snow moves from one raster cell to the next. This is perfectly captured by the *TPI*, making it a critical parameter for the model. This finding is supported by studies from Revuelto et al. (2020) and Meloche et al. (2022), pointing out the importance of the *TPI* for modeling snow distribution. All other parameters with higher importance are connected to wind characteristics and basic surface structures, which supports the significance of wind drifts and steep edges in the dynamics of snow distribution. However, it should also be noted that the used parameters only represent a small number of available parameters in relation to snow distribution.



Theoretically, the RF model is capable of taking a large number of parameters into account and still highlighting the most important ones. Our results also show that changing the input parameters impacts the performance of the model. Accordingly, future work should proceed in the same way, first using a large number of parameters and then successively limiting them to the most important ones. Based on our results, similar projects should consider significantly more parameters, generating better results by including metrics that capture wind and terrain exposure.

## 6 Conclusions

We present an analysis of snow distribution in palsa mires using a combination of field measurements, UAS LiDAR data, and RF-based calculations. This study provides significant insights into small-scale snow dynamics in palsa mires, revealing distinct patterns of snow accumulation at edges and cracks of the palsas due to wind effects and gravitational sliding. The increased snow depth provides thermal insulation, reducing the penetration of cold air during winter and resulting in degradation of frozen soil. Conversely, exposed tops of the palsas exhibit thinner snow cover, allowing deeper frost penetration but also longer exposition to solar radiation throughout the year.

Statistically, the RF model demonstrated a high predictive accuracy with a RMSE of 6.16 cm and an $R^2$ of 0.98, significantly outperforming the UAS LiDAR data, which had an RMSE of 26.73 cm and an $R^2$ of 0.59. The better performance underscores the effectiveness of incorporating parameters into the model, considering spatial wind- and terrain-related metrics. The *TPI* resulted as the most significant predictor of snow distribution, followed by parameters that consider the influence of wind like *Wind Effect* and *Valley Depth*.

Our results underscore the vulnerability of palsas to changing snow dynamics due to climate change. Increasing snow depth and altered wind patterns could intensify palsa degradation, leading to the loss of permafrost soils in Northern Finnish Lapland. Future research could expand this high-resolution snow distribution modeling approach to larger areas using satellite images, providing more comprehensive insights into the feedback mechanisms between snow cover, permafrost, and climate change. Our methodology can serve as a foundation for further modeling approaches, integrating knowledge about the importance of snow distribution for palsa development with other well-known drivers. It can be applied to other pan-Arctic palsa areas, continuous permafrost regions, and even for small scale avalanche forecasting in the Alps. Furthermore, general transferability in other earth surface phenomena is given, such as soil erosion or landform changes.

In conclusion, this study provides a detailed assessment of snow distribution within palsa mires and its implications for permafrost stability. The high accuracy of the RF model underscores the importance of incorporating spatial and environmental predictors in snow mapping, showing how small-scale dynamics can be unveiled to improve the understanding of permafrost evolution. Additionally, the results can be utilized in greenhouse gas measurements and footprint analyses, among other applications, highlighting their relevance for researchers focusing on the interactions between snow cover, permafrost, and greenhouse gas emissions.




**Appendix A**

**A1**

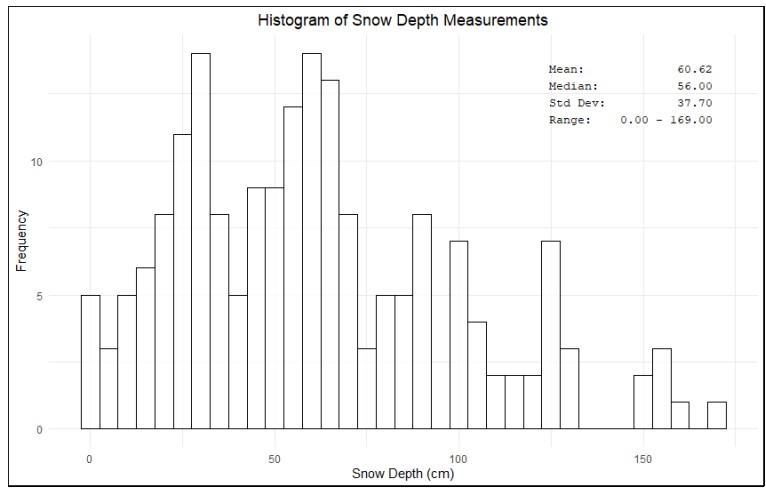

**Figure A1.** Histogram of all measured SD_{in-situ} points and respective statistics of the whole dataset.

*Code availability.* The R script used in this study is available upon request from the authors.

*Data availability.* Snow Depth and UAS data used in this study are available upon request from the authors. Meteorological data are available through the Finnish Meteorological Institute (https://en.ilmatieteenlaitos.fi/download-observations, FMI, 2024).

*Author contributions.* Initial study design: AS. Supervision: BB and TK. Data collection: AS, HS, MV, TK and PK. Data processing: AS, MV and PK. Data analysis and visualization of the results: AS, HS. Discussion of results and conclusions: AS, MV, BB and TK. Writing the
paper: AS, with contribution from BB, TK, MV and HS.

*Competing interests.* The authors declare that they have no conflict of interest.

*Financial support.* AS, BB and HS has been supported by the European Union's ERASMUS+ staff mobility programme and by the Graduate Academy of the Leibniz University Hannover. TK, MV and PK were supported by LANDMOD project (the Academy of Finland grant no. 330319) and CHARTER project (EU Horizon 2020 Research and Innovation Programme grant no. 869471).



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
