# Peer review of "Comparing High-Resolution Snow Mapping Approaches in Palsa Mires: UAS LiDAR vs. Machine Learning"

_EGUsphere, 2024_

## Author Comment (AC1)

**Point-by-point replies to the question and comments by Reviewer 1**

Dear Reviewer 1,

we are pleased to submit the replies to your questions and are thankful for the insightful comments and many good suggestions, as well as we are grateful for your time and effort in providing valuable feedback. We believe that addressing the issues raised by you, have now substantially improved our manuscript.

We hope our answers meet your approval. Your comments and our point-by-point responses are presented below. Please note, that we added a detailed description of a new RF modelling approach in the appendix A.

| Reviewer #1 comments | Action | Response |
|---|---|---|
| 1. The paper "Comparing High-Resolution Snow Mapping Approaches in Palsa Mires: UAS Lidar vs Machine Learning" by A. Störmer et al. aims to quantify the accuracy and efficiency of mapping snow depth over three palsas in northern Finland, in a spatially continuous raster-based map. Specifically, they choose two methods to compare: 1) using a Lidar sensor on a drone with two acquisition dates of data (no snow and snow), and 2) modelling snow depth based solely on a digital elevation model and using the machine learning algorithm "Random Forest". In situ data of snow depth are collected and used for training and validation. It is an interesting idea, and the need of mapping snow-depths over permafrost features is of great interest. It is also hard work, as noted by the authors in the Discussion, and the contribution of this paper will be of use for those wishing to map snow cover over terrain that has large variations over short distance, such as palsas. The conclusion was that the Random Forest model gave superior results as compared to the UAV Lidar. However, I have some major questions about the process and conclusions that must be addressed, as I question the overly optimistic result presented from the Random Forest model. The two larger issues to be addressed are below, followed by general and specific comments. | Answered | We appreciate your recognition of the relevance of our study and acknowledge the concerns raised regarding the optimism of the Random Forest model results. In response to your suggestions, we have conducted an additional model run incorporating the recommended adjustments. Specifically, we have removed vegetation from the initial DSM and applied hyperparameter tuning as well as cross-validation to the Random Forest model. The updated results, which provide a more robust evaluation, are presented in Appendix A. |
| **Larger issues that need to be addressed:** | Changed/ Answered | We have provided detailed answers to the two larger issues in Appendix A. |
| 2. Why was a Digital Surface Model and not a Digital Terrain Model used for calculating the ground in the no-snow data, and how does | | |

| | | | |
|---|---|---|---|
| | this affect the snow-depth measurements, and even the topographic derivatives used in the RF Model? | | |
| 3. | If the authors used cross-validation and present it as the accuracy of the model, then this result is over-optimistic and the comparison of UAV-Lidar to the Random Forest model result is biased and not a fair comparison to make. | | |
| 4. | 1 - Use of a DSM to represent ground level - It appears that the authors have made a Digital Surface Model (DSM) from the Lidar point data to represent the ground, rather than create a Digital Terrain Model (DTM) from the Lidar data. The DSM represents the height of all objects on the surface, and if there are shrubs on the palsas (which is typically the case in degraded palsas), they may be 35-50 cm tall. Therefore if a DSM was used to represent the ground in August, while insitu snow-depth measurements were taken from the ground up, the reported snow-depth will be highly affected by the height of the vegetation, and this will then vary over the whole surface of the palsa. If the authors have a reason for using a DSM rather than DTM, it is not clear in the article, and it needs to be motivated. Using a DSM will result in error in the snow depth measurements as presented. To create a DTM from your existing data is not difficult. If you look at the paper by Jacobs et al., 2021, you will see reference to papers that discuss the potential errors of snow depth measurements when DSMs are used.

In addition if the DSM was used to calculate the Topographic derivatives used as input parameters to the RF model, are these derivatives valid? | Changed/

Answered | See Appendix A. |
| 5. | 2- Cross validation - As I understand what has been done, the results of snow-depth for UAV Lidar and RF Modelling have been evaluated differently. In the case of UAV Lidar, the in situ data act as a fully independent data set used for calculating RMSE and the accuracy of the snow-depth measurements. In the case of the RF Modelling, the in situ data are used for training of the model, and the validation of the model as presented (see Fig 8) seems to have been made using a 10-fold cross-validation. In any case, the latter means that the data used to create the model are also used to evaluate the model. Cross-validation is never an assessment of the resulting map accuracy but is an assessment of the fit of the model. So it is no surprise that the authors get seemingly much better results for the RF | Changed/

Answered | See Appendix A. |

| | | | |
|---|---|---|---|
| | Model – the comparison is biased in the favor of the RF Model. Figure 8 shows this clearly, and to me is misleading. So the conclusion, as in the Results on Line 367/368, that the RF Model is showing its strength without high bias, I think is not valid.

The only way to fairly compare the assessments of these two would be to develop a model using in situ data from one palsa and apply the RF model developed to the other two palsas and assess the accuracy using the in situ data from those two palsas. Or, you could take insitu data from half of each palsa and developing training and accuracy datasets. (Note that if you consider taking a random selection of the insitu data for training/accuracy it is not optimal, since you will have spatial autocorrelation issues due to the proximity of the points, which is why the previous suggestions are better. ) | | |
| **Other general** | | | |
| 6. | The title: Rather than using the term "Machine Learning", I think it would be better to refer to this as "Modelling", because it doesn't make sense to me to compare it to the specific algorithm that is used, but rather that you have created a model to predict snow depth. | Changed | We agree with your suggestion to change the specific term "Machine Learning" to "Modelling". |
| 7. | There have been scientific articles that have mapped snow with UAV Lidar, eg, Jacobs, J.M. et al., 2021 "Snow depth mapping with unpiloted aerial system lidar observations: a case study in Durham, New Hampshire, United States" in The Cryosphere. (https://doi.org/10.5194/tc-15-1485-2021). While this may be the first paper to be published using UAV Lidar for snow on a palsa, I think that the Introduction should review and refer to articles that have generally applied UAV Lidar mapping of snow over other landscape types. | Changed | We acknowledge the importance of previous studies that have applied UAV LiDAR for snow depth mapping and appreciate your suggestion. Our initial focus was primarily on demonstrating the feasibility of snow depth modelling using Random Forest and assessing its implications for palsas. However, as the study evolved, the focus shifted more towards a comparative analysis between Random Forest modelling and LiDAR-based snow depth estimation. In response to your suggestion, we have incorporated additional references on UAV LiDAR-based snow depth mapping. Specifically, we adopted the vegetation removal approach inspired by Jacobs et al. (2021) and further reviewed relevant studies, including those by Avanzi et al. (2020) and Harder et al. (2020). |
| 8. | Section 2.1 is lacking a description of vegetation heights on the palsas. | Changed | We have added a description of the typical heights of common palsa vegetation. *Dwarf birches* are 15 and 60 cm (Betula nana) and *dwarf shrubs* 5 – 20 cm, while *sphagnum* |

| | | | |
|---|---|---|---|
| | | | *moss* (up to 3 cm) and *lichens* (<3 cm) are considerably smaller. These height estimates are based on our field observations. |
| **The following points all refer to Section 3.1 – Data collection** | | | |
| 9. Did you Post-Process the UAV Lidar data with RINEX data from a base station? If so, what was the base station (ie, source of the RINEX data)? | Changed/ Answered | | We used RINEX data for post-processing the LiDAR flight trajectory in POSPac. For both datasets, we obtained the RINEX data from the National Land Survey of Finland (NLS) CORS station in Kilpisjärvi (*KILP 2147250.4266 820562.0462 5930136.8831*). Further details can be found on the NLS website. This information has been incorporated into the newly added paragraph 3.1.1 (see comment #17). |
| 10. Parameters for the UAV flights are needed, eg, flying altitude, were cross-wise flights used? Knowing the directions of the flight lines is important because there are some Lidar measurements of 0 cm snow depth, and 50-60 cm snow depth in the insitu data, and it might be explained (possibly?) by not acquiring Lidar data in multiple angles – but I am not sure what has been done. | Changed/ Answered | | We have included additional details about the LiDAR flights in the newly added paragraph 3.1.1 (see comment #17). The flight altitude during the summer data collection was 30 m for each palsa, while in winter, the flight altitude was 60 m. All flights had a 50% side overlap. Cross-flights were not conducted, as we determined that they would not provide additional valuable data due to the specific environmental conditions (flat terrain with low vegetation). The flight direction was primarily along the longitudinal axis of the palsas, except for the summer flight over Peera palsa, which followed an east-west orientation. To improve clarity, we have included the flight trajectories in Figure 4 (see Appendix C). Additionally, we have provided point density values for each flight in $m^2$. The RGB flights were conducted using an Autel EVO II Pro V2 UAV at a flight altitude of 80 m, with a 75% side overlap for each flight. Initially, we had stated that the orthophotos were acquired with the integrated RGB sensor of the LiDAR mapper. However, this was a misunderstanding, and we have now corrected this statement. The orthophotos do not contribute specific data to the analysis but were solely used for figure creation. Regarding the occurrences of 0 cm snow depth in the UAS-LiDAR data: These values |

| | | result from our initial processing step, in which we set all negative values in the computed DSM to zero. This aspect is now explicitly mentioned in Appendix A, and we have adjusted this approach in the updated modelling. |
|---|---|---|
| 11. Line 151/152 says that GCPs were set out. Was this for both the Lidar and the RGB images? How many GCPs? And then, what was the horizontal and vertical accuracy of your data – both the Lidar and the RGB images? | Changed/

Answered | For all winter flights (LiDAR and RGB) and palsa sites, four GCPs were used, positioned around the palsa. The accuracy for each GCP is between 1–2 cm.

We have established several permanent GCPs (measured with RTK-GPS) located on known points of large stones in the study sites. Permanent GCPs have been established because we are monitoring changes in the palsas by collecting drone data annually since past 8 years. The accuracy of these RTK-GPS-measured GCPs is between 1–2 cm. For all UAS-LiDAR summer flights we utilized these GCP's: three for Peera, 20 for Pousu, and 30 for Puolikkoniva. The RGB flights were conducted using the drone's internal RTK system. However, we consider the accuracy of these flights to be not relevant in the context of this study, as the orthomosaics are used solely for overview purposes.

We have added this information to section 3.1.1. |
| 12. Line 153 – Change orthopictures to images, since the raw images are not orthorectified yet. That's a later step. | Changed | We agree and changed the term. |
| 13. Line 157/158 "Structure from Motion techniques were not applied…" I do not understand why this sentence is here. If you created an orthophoto, which you say you do in the next sentence, then you have applied photogrammetric image matching (how you define SfM and if you define it differently than photogrammetric image matching determines what term you like to use). But why even say what you haven't done? State what you have done to produce the orthophoto. | Changed | We acknowledge the potential for misunderstanding and have clarified our statement in lines 157/158. Specifically, we used SfM techniques solely for the creation of orthophotos. As noted in comment #10, there was an initial misunderstanding regarding the generation of the orthophotos. We have now revised this section to ensure clarity and accuracy. |
| 14. Line 164 – I think you mean snow depth rather than snow cover. | Changed | We agree and changed the term. |
| 15. Line 166 – RTK-GPS. | Changed | We agree and added "GPS". |
| 16. It says on line 173 that there are randomized points on the edges of Puolikkoniva, but I do not see very many of these (maybe 5 at | Changed/ | In Puolikkoniva, 20 randomized points are located at the edges, which corresponds to one fifth of the total point dataset for this |

| | | |
|---|---|---|
| most?). In hindsight, I would guess that you would want to have made cross-wise transects on this palsa. Take this up in the Discussion if so. | Answered | palsa. We consider this distribution to be adequate.

To provide a clearer visualization of the point distribution, we have added an additional figure to the appendix, categorizing the points by classes. In this figure, we have also included an orthophoto without snow cover to enhance the visibility of point distribution in the most extreme areas. Furthermore, we have adjusted the point style to improve differentiation and recognition (Appendix E). |
| **Reference (in situ) data** | | |
| 17. I think you need a separate section to describe Reference data collection – either two sub-sections under 3.1 or else 3.1 for UAS data collection and 3.2 for Reference data collection. Under the reference data collection, there should be a better description regarding how the insitu snow depth measurements were made, specifically, was the GPS Z-measurement made from the ground level? Was it a yardstick, and was a level used to make sure it was normal to the surface? | Changed/

Answered | We have revised the manuscript to create a clearer structure for data collection. In particular, we have introduced two separate subsections: 3.1.1 for data collection at universities of applied sciences and 3.1.2 for the collection of reference data.

In response to your suggestion, we have expanded the description of the reference data collection. The snow depth measurements were carried out with a heavy wooden yardstick. GPS-Z measurements were not taken from the ground, as this approach is subject to large uncertainties. In addition, no level was used to ensure that the measurements were perpendicular to the surface. These methodological details were explicitly mentioned in the revised section to improve transparency and reproducibility. |
| 18. For the insitu data you need at some point to say that these also may have errors and what these errors may be caused by, and how they may affect your result. Since the RF model is completely based on the insitu data, the errors of the insitu data are simply propagated, but do not affect the evaluation. For validating the Lidar data derived snow depths, the potential measurement errors of the insitu data are only accounted for in the evaluation. | Changed/

Answered | We have expanded our discussion on the uncertainties associated with snow depth measurements in lines 409–411. In this revision, we have explicitly acknowledged that measurement errors not only influence the modelling results but also affect their statistical evaluation. We have critically assessed the extent of this impact.

However, we still consider that manual snow depth measurement is the most accurate method available, with only minor deviations. Given the overall reliability of this approach, we do not expect these uncertainties to significantly affect the conclusions of the study. |

| | | | |
|---|---|---|---|
| 19. | Also, think about whether the section on UAS data collection is only about data collection or if you want to describe the processing of the data here – in which case you might just name it "UAS data" or "UAS data collection and processing". | Changed | We agree with your suggestion and have renamed section 3.1.1 to "UAS Data Collection and Processing" to better reflect its content. As noted in previous comments, we have also expanded this section to include additional details on both the data collection and processing procedures. |
| 20. | The in situ data particularly in the case of the largest palsa Puolikkoniva were run in two transects lengthwise along the palsa, but not crosswise, over the edges where the deepest accumulation of snow may have been. Therefore the values where some of the largest differences are between the Lidar and the RF Model cannot really be assessed, making the assessment incomplete – the shortcoming must be acknowledged. | Answered | We respectfully disagree with this point. As discussed in comment #28, we believe that our dataset is well-suited for the objectives of this study. Our measurement strategy included data points in key areas of the palsa, such as the summit, steep slopes, and internal trenches, ensuring that the primary variations in snow depth were captured.

However, we acknowledge that extreme values may still have been missed due to the inherent limitations of in situ snow depth measurements. We have addressed this explicitly in the discussion part about limitations. |
| 21. | Also the Lidar may measure extremes in snow-depths, while the model will not if it does not have representative data for the extremes. Therefore there will be more variability in the Lidar data, but we cannot tell which is "wrong". | Answered | As mentioned in response to the previous comment, we believe that our dataset adequately captures most of the extreme snow depth variations. However, we acknowledge that some extreme values may not have been fully represented in the model due to limitations in the in-situ data collection. Therefore, we address this issue in the limitations section and emphasize the differences in variability between LiDAR measurements and the RF model. |
| **Section 3.2 – RF algorithm** | | | |
| 22. | The authors state on Line 189 that no explicit hyperparameters were specified. So this means that they were not analyzed, although the outcome of the model is what is being assessed as the main objective of the article. It is not difficult to assess the hyperparameters using Grid-Search or another comparable function. | Changed/

Answered | See Appendix A. |
| 23. | Permutation mode was used for variable importance – do you know how this works? Is it a single run of the RF model? When you run PI repeatedly, do the same variables have the same importance? The random nature of RF often requires running variable importance (or in this case PI) many times (eg, 100) and taking an average. Even then, one needs to be | Changed/

Answered | See Appendix A. |

| | | careful with their interpretation of variable importance. | | |
|---|---|---|
| 24. For Line 187-188 - I'm not really sure what you have done with the model and the in situ data. You state that you have split 70% training and 30% test. Is this used by RF for internal cross-validation of the model (if you split the data 70/30 in the RF model, then it is likely this is how it is being used). Is this done with replacement? If you have removed 30% of the data for independent evaluation, then you need to clearly state this, but I don't think this is what you have done. | Changed/ Answered | We separated the dataset into 70% training data, which was used for the Random Forest model, and 30% test data. The test dataset was not utilized for internal cross-validation but was exclusively used to compute performance metrics, including RMSE, $R^2$, MAE, and standard deviation. Further details on this methodology are provided in Appendix A. |
| 25. Line 184 – The dependent variable for your model is snow-depth. | Changed | We agree, and to clarify this, we have removed the term "dataset" in line 184. |
| 26. Line 185 –"Input parameters" are mentioned here but we don't know what they are until later. Couldn't you refer to Table 2 here? Otherwise we are left wondering what the parameters are. | Changed | We agree and added Table 2 as reference. |
| 27. Line 189 – delete "precise" – This is a judgmental word – leave it to your results to be the judge of that. | Changed | We agree and removed the word *precise*. |
| 28. In addition, RF models are sensitive to imbalance in the training data, and also do not extrapolate beyond the minimum and maximum snow-depth values (or whatever the target variable may be). How are your results affected by this, and how might others in the future be affected by this and what would your recommendations be to future applications of this method? | Answered | We acknowledge the sensitivity of Random Forest models to imbalances in the training data, as well as their inability to extrapolate beyond the observed minimum and maximum values of the target variable. To ensure the robustness of the model, it is crucial to capture a dataset that adequately represents the full range of snow depths within the study area.

A thorough understanding of the investigation area is essential, and snow depth measurements should specifically target extreme locations, such as exposed areas on the palsa summit and accumulation zones along the edges. However, conducting such measurements in these environmental conditions is both time-consuming and labour-intensive.

For future applications of this method, we recommend careful planning to ensure representative data collection. This includes identifying extreme locations in advance by analysing orthophotos and previously acquired digital elevation models (DEMs) to optimize the placement of measurement points. We have added this to our discussion. |

| | | |
|---|---|---|
| **Section 3.3 –** | | |
| 29. The first sentence needs rewriting. First of all, which "collected airborne data" is referred to here? I assume it was the August DSM from Lidar that was used? It is not stated. Were these data processed differently than what was described in Section 3.1? Declare which DEM you are working with and say specifically that you are creating parameters from this. What happens if you use a DSM and create all of these topographic derivatives as parameters? Are those new derivatives valid, such as Topographic Wetness Index, if they are based on the surface elevation which includes vegetation? This must be well-motivated if the authors believe that there is a valid reason for this. | Changed/

Answered | We acknowledge the need for greater clarity in the first sentence and have specified that we are referring to the summer dataset. To improve transparency, we have moved this information to the newly added section on UAS-LiDAR processing (Section 3.1.1).

Regarding the influence of vegetation on the derived topographic parameters, we have addressed this aspect in previous comments and provided further details in Appendix A. |
| 30. Line 210 – If a 0.3 m buffer was used were the values for any parameters averaged within this area? | Answered | The input parameters were not averaged within the 0.3 m buffer areas. Instead, each input parameter value was directly linked to the corresponding in-situ snow depth ($SD_{in\text{-}situ}$) measurement. Consequently, each snow depth value is associated with an average of 28 input parameter values.

For Puolikkoniva, 100 snow depth measurements were linked to a total of 2,819 input parameter observations, while for Peera and Pousu, the corresponding values were 39 to 1,097 and 46 to 1,306, respectively. In total, 5,222 points were generated for the Random Forest model, with 70% used for training and 30% for validation. This approach was chosen to reduce noise and smooth the resulting dataset.

You can find further information in Appendix A. |
| 31. Table 2 – 12 parameters were used, but 21 are in the table. Could you indicate in a way what parameters were used? | Changed | We have described the input parameters used for each model run in lines 205–209.

However, based on your suggestions and the revised model design, we have decided to focus on a single model run. As a result, we used only 12 parameters and removed the information related to previously unused parameters and former model runs. The section on input parameters and model runs has been updated accordingly. |
| 32. For the Discussion: When you made the insitu measurements, it was August, and the palsa | Changed/ | We have already incorporated the findings of Renette et al. (2024) into our discussion (lines |

| | | | |
|---|---|---|
| | had likely subsided. Renette et al., 2024 show that the difference between elevation in September (likely maximum thaw depth of the Active Layer) and April (minimum thaw) was on average 15 cm, and up to 30 cm in some areas, albeit on a taller palsa than in the study presented here. In any case, this may mean that trying to measure snow depth using a DTM from September may introduce errors if the terrain is actually elevated some cm more than this. This is hard issue to solve with UAV Lidar, since you would need to be in place to create a DTM right after snow-melt, and all snow would need to have melted. So, you need to discuss what implications this has to your results. Also, since you have RTK-GPS data, and you have measured to the ground I assume, you actually have a dataset where you could compare the Z-measurement from March to the DTM from August, and get an estimate of the difference in height between the max-thaw and min-thaw state of the palsa. | Answered | 384–387). However, based on your comments, we have expanded this discussion to explicitly address the potential impact of seasonal elevation changes on the accuracy of $SD_{LiDAR}$ measurements. In contrast, the $SD_{RF}$ results should be less affected, as the modelled snow depth is independent of seasonal elevation fluctuations.

Furthermore, we have considered this aspect when refining our overall evaluation of the comparison between RF and LiDAR results. As you pointed out, accurately capturing snow depth using LiDAR is only possible if data collection occurs immediately after snowmelt, once all snow has disappeared.

Regarding the RTK-GPS measurements, we did not measure directly to the ground. During fieldwork, we observed that the thick and frozen vegetation layer made it challenging to reach the true ground surface using the RTK stick. Instead, we found that the fine yardstick provided a more accurate way to measure snow depth. Consequently, we are unable to compare Z-measurements within our datasets. However, we have incorporated this consideration into our discussion of future research implications (lines 386–387). |

**Language**

| | | | |
|---|---|---|
| 33. | It's my feeling that some value judgement words don't belong in a scientific article. Such as "exemplarily" on line 53. | Changed | We reviewed the manuscript for judgemental words and changed these accordingly. |
| 34. | Line 38 – deepening instead of growth. Line 58 – deeper instead of higher. | Changed | We changed these words. |
| 35. | Otherwise some minor grammatical fixes once the paper is revised can be looked over. | Changed | We will have a final grammar check after implementing all changes to the manuscript. |

**Specific**

| | | | |
|---|---|---|
| 36. | Line 35 – it is not only bound by peatland presence but also climatic parameters | Changed | We agree and added "and driven by climatic parameters". |
| 37. | Line 69 – "Satellite data" only names the platform. What kind of satellite data are you referring to? Optical? Radar? That is the more important aspect. Similar issue is on line 74 where the sensor type should be mentioned and not just the platform which is UAS/UAV. Look through your paper for these kind of omissions. | Changed | We acknowledge the need for greater specificity regarding the types of satellite data referenced. In the respective sections, we now explicitly state that we are referring to both optical and radar satellite data. |

| | | | Additionally, we have specified the type of UAS sensor used in each mentioned study to ensure clarity. These adjustments have been implemented in lines 9, 14, 49, 71, 74, 75, and 95. |
|---|---|---|---|
| 38. Line 70 – change technical limitations to properties | Changed | | We agree and changed the term. |
| 39. Line 86 – the authors mention 3 methods, but the title takes up two. The third method seems to be the insitu data, but that has been used to train the RF Model, and I don't think you are really assessing the accuracy of the method, so I would stick to the two methods. | Changed | | Thank you for highlighting this inconsistency. We have revised our focus to explicitly center on the two primary methods - LiDAR-based snow depth estimation and RF modelling. Accordingly, we have adjusted the structure of our objectives and intentions. |
| 40. Line 89 – delete simulation. You are just modelling. | Changed | | We agree and deleted "simulation". |
| 41. Table 1 – the photos are rather small. Can they be made bigger. Put the date (day-month-year) of the photos in the Table text. | Changed | | We agree and changed the caption and increased the size of the images. |
| 42. Line 129 – For what year or years is that the annual mean temperature? | Changed | | We inserted "For the time period 1991 – 2020, …". |
| 43. Line 137 – For what location is that the duration of permanent snow cover? | Changed | | This value is specific to Kilpisjärvi, and we have incorporated it into the text. |
| 44. Figure 2 – What is shown in Fig 2? It needs to be said clearly in the Fig text. Is this an average value for 1990-2020? It would be very helpful to know what the climate conditions were for the years in which you acquired the snow data. Was it a very snowy year? Windy in the days before you visited? Warm temperatures so that the snow melted some? Knowing these conditions can help us to explain any differences between the various results, particularly if the model is solely based on the DEM. I see you mention this on Line 401/402. | Changed/ Answered | | Figure 2 presents the average monthly snow depth (cm), temperature (°C), and precipitation (mm) recorded at the Kilpisjärvi weather station for the period 1990–2020. This timeframe was selected to align with the 30-year reference period established by the World Meteorological Organization (WMO). However, we recognized that the appropriate reference period should be 1991–2020 and have updated the figure accordingly (Appendix B).

The purpose of this figure is to provide a general overview of the climatic conditions in the study area. Since snow depth data were collected on only two days (March 23–24, 2023) under stable weather conditions, we do not believe that presenting weather data from the preceding days or the entire winter season of 2022/23 would provide additional meaningful insights.

To clarify this, we have explicitly stated in line 141 that all snow depth measurements were conducted on March 23–24, 2023. |

| | | |
|---|---|---|
| 45. Line 141 – Write which day the data were acquired. If you cannot fit it reasonably in the text, because it was different dates for different palsas, I suggest you put it in Table 1 – dates for image and Lidar acquisition. | Changed | We have added the specific dates of data acquisition in line 141. The UAS-LiDAR data were collected on August 27, 2022 (summer) and March 23, 2023 (winter). Snow depth measurements were conducted on March 23, 2023, for Puolikkoniva and Pousu, and on March 24, 2023, for Peera.

Additionally, we have included the exact dates of LiDAR data acquisition in Table 1 to ensure clarity and consistency. |
| 46. Several of the Figures have such small text that they are difficult to read. Eg Fig 3. | Changed | We have increased the font size for Figures 3, 6, 7 and 8. |
| 47. Section 3 – Is August the season for maximum thaw? It's not September? Does Verdonen et al. 2023 state that August is the max ALT? If it is August, I think you should more specifically say the end of August. If you aren't sure or don't have a reference to back it up, then maybe it is more reasonable to say that the end of August is near max ALT. | Changed | We agree that this statement requires greater accuracy. We have revised it to indicate that the maximum ALT is typically reached between the end of August and mid-September, depending on annual weather conditions and the onset of the freezing season. |
| 48. Line 231 – 240 feel like they belong in the section describing the RF model. | Changed | We agree and moved this part to the description of the RF algorithm and modelling data preparation. |
| 49. Line 231/232 – Was the 10-fold cross-validation done when creating the initial RF model, or was this something that was done afterwards and used as the "validation" data presented in Figure 8? If it is the latter, you cannot say that it was used to reduce over-fitting in the model? There is an option in Random Forest to use cross-validation to create the model, and that is one tool of several to reduce over-fitting. Other ways to reduce over-fitting is to limit tree depth, -- by the way, in Section 3.2 you mention target node depth, but I don't see in the caret package what that refers to. Is it "maxdepth"? In that case I suggest you name the parameter in parentheses. | Changed/

Answered | See Appendix A. |
| 50. Line 236/237 – What are "the initially calculated values"? You are using the insitu data to train a RF model and then evaluating the model based on a cross-validation that using that same insitu data. See my point #2 under "Larger issues". | Changed/

Answered | See Appendix A. |
| 51. Line 273/274 – "Only a few narrow structures with significantly higher snow can be recognized based on the UAS LiDAR data" – I do not know what this sentence is about. | Changed | Our intention was to highlight that only small areas within the study region exhibit significantly higher snow depths in the UAS-LiDAR dataset. To clarify this, we have revised the sentence as follows: |

| | | | “On the other hand, only small areas with significantly higher snow depth in the UAS-LiDAR dataset compared to the RF dataset are detectable in certain regions surrounding the palsas.” |
|---|---|---|---|
| 52. Line 281 and Fig 7 and Table 3 – I don't think we need to see all 3 model runs, just the best one. | | Changed | We agree and changed the text, figure and table accordingly. See comment #31 and Appendix A. |
| 53. Line 285 – rather confusing that it is stated that Elevation was removed, and now it is important. Also Fig 7 text is impossible to read because it is so small. | | Changed/ Answered | Please refer to our previous responses regarding the removal of the initial model runs. As a result, it is no longer necessary to elaborate on the exclusion and reintroduction of the *Elevation* parameter.

For clarity regarding our initial approach: *Elevation* was excluded in the second model run because all other input parameters were derived from it. This step was taken to assess whether *Elevation* might introduce bias into the modelling results. After analysing the outcomes, we found no indication of such bias and subsequently decided to retain *Elevation* as an input parameter in the final model. |
| 54. Line 295 and Table 4 – these areas of "Top", etc, could you have a figure somewhere – maybe supplemental where these areas are shown? Do we know the number of samples (n) in each group? | | Changed/ Answered | See Appendix A and E. |
| 55. Line 323 also Line 346 – Fig 9? | | Changed | Thank you for the note, we have changed that. |
| 56. Figure 9 – Is B (Slope in degrees) based on the DSM? Is this valid then to calculated slope based on vegetation? | | Changed | See Appendix A. |
| 57. Line 404/405 – I guess you are referring to reflectance of the lidar from the snow/ice surface? If so I think you should have a reference here. | | Changed | We agree and have added a reference to Deems et al. (2013), which investigates the influence of reflectance and scattering by snow and ice surfaces on the accuracy of LiDAR sensors. |

**Appendix**

In this section, we provide additional information addressing comments #4, 5, 10, 16, 22, 23, 24, 44, 49, 50, 54, 56 from Reviewer 1 and #5, 6, 8, 10, 38, 40, 41, 52, 53, 63, 66 from Reviewer 2.

We sincerely appreciate your insightful comments and suggestions, which have significantly contributed to improving both the modelling approach and the overall quality of the manuscript.

**Appendix A**

To ensure high-quality modelling results and accurate snow depth distribution maps derived from UAS-LiDAR, we implemented your recommendations, including the removal of vegetation from the LiDAR-derived products and a re-evaluation of the modelling approach.

Additionally, we incorporated hyperparameter tuning and cross-validation to determine the most suitable parameter settings for the Random Forest model. To further improve model robustness and prevent overfitting, we also adjusted the data splitting strategy by testing the RF model on an independent external dataset.

1.  Removal of vegetation from UAS-LiDAR DSM

Our initial decision to retain vegetation in the modelling process assumed that small and dense vegetation, as present in our study sites, is difficult to remove - even from point clouds. Testing several vegetation filter algorithms, such as the *Cloth Simulation Filter* (CSF) and *Statistical Outlier Removal* (SOR) in CloudCompare, confirmed this assumption, as the vegetation was not properly removed in the resulting products.

Additionally, we considered that vegetation significantly influences snow depth distribution by enhancing snow retention capacity. Therefore, we initially decided to include vegetation in the modelling process, expecting it to be beneficial for RF modelling.

However, based on your suggestions, we tested the *Progressive Morphological Filter* (PMF) Algorithm as described by Zhang et al. (2003) and Jacobs et al. (2021) and obtained satisfactory results with an effective removal of vegetation. We applied PMF filtering using the following parameters:

*   Window sizes: 0.5, 1, 2, and 3
*   Thresholds: 0.05, 0.1, 0.3, and 0.5

The extracted ground and vegetation points were saved in point cloud format. Using CloudCompare, we generated a DTM for each palsa using the Rasterize function. Empty cells within the point clouds were interpolated with a triangle max edge length value of 5.0.

The newly created DTMs were then used to recalculate the snow depth distribution for all three test sites in GIS, following the methodology described in the manuscript. In our initial calculations, all negative values were set to zero. However, in this revised approach, we retained negative values to highlight areas where either the LiDAR sensor produced inaccuracies or surface degradation occurred between the summer and winter flights.

Based on these refined DTMs, we recalculated all input parameters used in the final RF model run in SAGA GIS. The following 12 parameters were included: *Aspect, Elevation, Channel Network Base Level, Channel Network Distance, Negative Openness, Positive Openness, Relative Slope Position, Slope, Topographic Position Index, Valley Depth, Wind Effect, Wind Exposition*.
A detailed description of these parameters is provided in Table 2. We have now focused on a single model run, and accordingly, we have removed descriptions of other parameters from the manuscript to ensure clarity and consistency.

2. Splitting data into training and test datasets

In the initial study design, we used the entire buffered $SD_{in-situ}$ dataset to extract the input parameters from the raster stack, resulting in a data frame with 5222 points. We then split this dataset into 70% training and 30% test data. However, this approach introduced a risk of overfitting, as each $SD_{in-situ}$ point was represented an average of 28 times in the dataset. Consequently, many points appeared in both the training and test datasets, reducing the independence of the validation process.

To address this issue, we revised our study design by first separating 70% of the point features from each $SD_{in-situ}$ dataset for training and 30% for testing. Only after this separation did we extract the input parameter values for the training dataset, ensuring a clear distinction between training and validation data. The test dataset was reserved exclusively for model validation. The following extract from the R script illustrates these steps:

```
**===== Function to split training and test dataset =====**
split_shapefile <- function(shp) {
  set.seed(42)  # Ensure reproducibility
  num_samples <- nrow(shp)  # Get the number of samples
  train_indices <- sample(num_samples, size = round(0.7 * num_samples))  # Select 70% of the samples for training
  test_indices <- setdiff(1:num_samples, train_indices)  # The remaining 30% for testing

  shp_train <- shp[train_indices, ]  # Create training dataset
  shp_test <- shp[test_indices, ]  # Create test dataset

  return(list(train = shp_train, test = shp_test))  # Return the split datasets as a list
}

**Splitting the dataset for all three locations**
split_pousu <- split_shapefile(shp_pousu)
split_peera <- split_shapefile(shp_peera)
split_puolikkoniva <- split_shapefile(shp_puolikkoniva)

**Combine training and test datasets for all Palsas**
shp_train_all <- rbind(split_pousu$train, split_peera$train, split_puolikkoniva$train)  # Merge training datasets
shp_test_all <- rbind(split_pousu$test, split_peera$test, split_puolikkoniva$test)  # Merge test datasets
```

After extracting the input parameters from the raster stack, the final dataset consisted of:

- Training dataset: 3,645 points (Puolikkoniva: 1,983; Pousu: 905; Peera: 757)
- Test dataset: 1,577 points (Puolikkoniva: 836; Pousu: 401; Peera: 340)

To prevent errors and miscalculations, all NoData values were removed from the datasets, resulting in a final training dataset of 3,504 points and a final test dataset of 1,548 points for further modelling and validation.

3. Hyperparameter tuning and cross validation

To determine the optimal values for *mtry*, *min.node.size*, and *sample fraction*, we performed hyperparameter tuning using the *mlr* package in R (Bischl et al., 2016).

To prevent overfitting, we restricted the search range for *min.node.size* to 10–15 and for *sample fraction* to 0.7–0.85, following the recommendations of Probst et al. (2019) and Breiman (2001). Allowing an unlimited search range initially resulted in better model performance, but at the cost of reduced generalization, indicating signs of overfitting. We selected the final search range based on multiple test runs with different settings.

For cross-validation, we tested different fold sizes to identify the most effective configuration. The best results were achieved using a 4-fold cross-validation. The following R script extract provides details on the tuning process:

```
**===== Hyperparameter Tuning with tuneRanger (Regression) =====**

**Define the regression task**
task <- makeRegrTask(data = all_train, target = "Class")

**Define the cross-validation strategy**
cv_desc <- makeResampleDesc("CV", iters = 4)  # 4-fold cross-validation

**Define the Random Forest learner with hyperparameters as tuning options**
learner <- makeLearner("regr.ranger", num.trees = 1000)

**Define the hyperparameter search space**
param_set <- makeParamSet(
  makeIntegerParam("mtry", lower = 2, upper = ncol(all_train) - 1),  # Number of variables to consider at each split
  makeIntegerParam("min.node.size", lower = 10, upper = 15),  # Minimum number of observations per node
  makeNumericParam("sample.fraction", lower = 0.7, upper = 0.85)  # Proportion of samples used in each tree
)

**Define the tuning control (e.g., Bayesian optimization or random search)**
control <- makeTuneControlRandom(maxit = 70)  # 70 iterations for tuning

**Hyperparameter tuning with cross-validation**
tuned_params <- tuneParams(
  learner = learner,
  task = task,
  resampling = cv_desc,  # 4-fold CV
  par.set = param_set,
  control = control,
  measures = rmse  # Root Mean Squared Error as the performance metric
)

**Display results**
print(tuned_params)

**Best Random Forest model with tuned parameters**
best_learner <- setHyperPars(learner, par.vals = tuned_params$x)
```

The final tuned hyperparameters were as follows:

- mtry: 9
- min.node.size: 10
- sample fraction: 0.79

4. Permutation Importance (PI)

In our initial study design, we conducted the RF modelling once and directly used the permutation importance (PI) values provided by the model.

In our revised approach, we refined this process by repeating the calculation 100 times to obtain a mean PI value for each input parameter, ensuring more robust and reliable importance rankings.

The following R script extract details the implementation of this approach:

```
**========== Permutation Importance**
num_repeats <- 100
importance_values <- matrix(NA, nrow = num_repeats, ncol = ncol(all_train) - 1)

for (i in 1:num_repeats) {
  cat("Iteration:", i, "\n")

  # Train the model using the identical hyperparameters from tuning
  temp_model <- ranger(
    x = all_train[, -ncol(all_train)],
    y = all_train$Class,
    mtry = tuned_params$x$mtry,  # Optimized mtry value
    min.node.size = tuned_params$x$min.node.size,  # Optimized min.node.size
    sample.fraction = tuned_params$x$sample.fraction,  # Optimized sample.fraction
    num.trees = 1000,
    importance = "permutation",
    seed = i  # Different seed per run for robustness
  )

  # Store the feature importances in the matrix
  importance_values[i, ] <- importance(temp_model)
}

**Compute the mean Permutation Importance over the 100 runs**
mean_importance <- colMeans(importance_values)
```

We modified Figure 7 to display only the 12 selected parameters along with their respective mean PI values over 100 iterations. Additionally, we normalized the values, setting the most important parameter (Topographic Position Index) to 1.

[Figure]

*Figure 7. Overview of normalized mean Permutation Importance values from RF modelling over 100 iterations.*

5.   Final results and validation

Both the RF-based and UAS-LiDAR-based results were validated using the initially separated test dataset. Additionally, we conducted three further RF model runs, where in each iteration, two palsa sites were used as the training dataset, and one was used as the test dataset. This approach further validated the generalization capability of the model.

The validation results indicate that the RF-based approach now exhibits lower peak accuracies compared to the initial study design. However, by reducing overfitting, the results are more plausible and robust, while still achieving high accuracy and outperforming the UAS-LiDAR-based approach:

Table 3. Overview of the calculated Root Mean Square Error (RMSE), Coefficient of Determination (R2), Mean Absolute Error (MAE) and Standard Deviation (SD) for RF- and UAS-LiDAR-based snow depth estimations. Additionally, external validation results (RMSE and $R^2$) for RF-based snow depth at each palsa site (Peera RF, Pousu RF, Puolikkoniva RF) are provided.

| Parameter | RF | LiDAR UAS | Peera RF | Pousu RF | Puolikkoniva RF |
|---|---|---|---|---|---|
| RMSE | 18.33 | 23.49 | 16.67 | 21.31 | 27.13 |
| $R^2$ | 0.77 | 0.691 | 0.628 | 0.767 | 0.578 |
| MAE | 13.26 | 17.49 | - | - | - |
| SD | 18.11 | 20.84 | - | - | - |

We recalculated all metrics for different point groups and included the number of points per group. These groups were classified visually, based on orthophotos, slope data, and elevation characteristics of the respective locations.

The results show that the accuracy differences between RF and UAS-LiDAR-based approaches are now less pronounced. However, in certain categories, such as *Thermokarst* and *Open Area*, the UAS-LiDAR-based results show lower accuracy, likely due to measurement inaccuracies caused by water surfaces and irregularities in areas with higher vegetation.

Table 4. Overview of RMSE, $R^2$, MAE and SD divided by validation point locations within the investigation areas.

| | RMSE | | $R^2$ | | MAE | | SD | |
|---|---|---|---|---|---|---|---|---|
| | RF | LiDAR | RF | LiDAR | RF | LiDAR | RF | LiDAR |
| On Top (n = 69) | 8.33 | 8.33 | 0.841 | 0.730 | 3.84 | 3.84 | 8.32 | 10.83 |
| Edge (n = 66) | 13.12 | 13.12 | 0.894 | 0.768 | 5.85 | 5.85 | 12.82 | 19.09 |
| Thermokarst (n = 16) | 10.99 | 33.73 | 0.893 | 0.592 | 5.42 | 30.35 | 10.69 | 25.08 |
| Open Area (n = 26) | 4.54 | 14.23 | 0.926 | 0.519 | 1.56 | 9.84 | 4.40 | 12.59 |

Figures 5, 6, 8, and 9 have been updated based on the new results.

Figure 5 now includes the recalculated snow depth maps. We have incorporated all areas where $SD_{LiDAR}$ values are below 0, visualizing these parts in red to highlight regions where the LiDAR sensor may have measured incorrectly or where degradation has occurred between flights.

[Figure]

*Figure 5. Snow depth predictions based on the RF model (left) and the UAS-LiDAR (right) at site Puolikkoniva (a, b), Pousu (c, d) and Peera (e, f) palsas. Red points are showing the in-situ snow depth measurement locations.*

In Figure 6 we inserted the new calculated difference maps and we also included the parts with negative values in red:

[Figure]

*Figure 6. Snow depth differences between modelled and UAS LiDAR results at a) Puolikkoniva, b) Pousu and c) Peera palsas.*

Figure 8 shows the scatter plots based on the 30% test dataset. Here we used only the single values of the SD$_{in\text{-}situ}$, not considering the values within the buffer areas of the test data. We decided to do it like that, to obtain a very fine validation of both methods:

[Figure]

*Figure 8. Scatter plots with regression lines for UAS-LiDAR-derived and RF-modelled snow depths, based on the external test dataset.*

Figure 9 has been updated to reflect the new results. Additionally, we have incorporated the calculated slope derived from the DTM of Pousu palsa.

[Figure]

a) Pousu palsa site and the analysis area (red).

b) Slope in degrees.

c) Differences in snow depth between RF and LiDAR UAS approach.

d) Snow distribution based on RF approach and highlighted cooling and warming locations.

e) Possible warming/cooling spots based on snow distribution and areas with existing or higher possibility of block erosion.

*Figure 9. Explanation of differences between UAS LiDAR-derived and RF-modelled snow depths.*

**Appendix B**

[Figure]

*Figure 2. Climate chart of Kilpisjärvi (FMI, 2022). Dotted line shows 2 m above ground temperature in °C, dashed line shows precipitation in mm and solid line shows snow depth in cm.*

**Appendix C**

[Figure]

*Figure 4. Snow depth measuring points within the investigation sites at Puolikkoniva (a), Pousu (b) and Peera (c) palsa illustrating different methods for recording snow depth (transects, randomized, crossed).*

**Appendix D**

[Figure]

*Figure A1. Histogram of $SD_{in\text{-}situ}$ points and respective statistics per palsa site.*

**Appendix E**

[Figure]

*Figure A2. Overview of classification of all SD$_{in\text{-}situ}$ points into classes Edge, On Top, Open Area and Thermokarst.*

**Appendix F**

*Table A3. Correlation between each input parameter and RF-modelled snow depth.*

| Parameter | Correlation to SD$_{RF}$ | Parameter | Correlation to SD$_{RF}$ |
|---|---|---|---|
| Aspect | 0.09 | Relative Slope Position | -0.49 |
| Elevation | -0.12 | Slope | 0.08 |
| Channel Network Base Level | -0.09 | Topographic Position Index | -0.87 |
| Channel Network Distance | -0.45 | Valley Depth | 0.50 |
| Negative Openness | 0.22 | Wind Effect | -0.55 |
| Positive Openness | -0.50 | Wind Exposition | -0.80 |

**References**

Avanzi, F., Zheng, Z., Coogan, A., Rice, R., Akella, R., and Conklin, M. H.: Gap-filling snow-depth time-series with Kalman Filtering-Smoothing and Expectation Maximization: Proof of concept using spatially dense wireless-sensor-network data, Cold Reg Sci Technol, 175, https://doi.org/10.1016/j.coldregions.2020.103066, 2020.

Bischl, B., Lang, M., Kotthoff, L., Schiffner, J., Richter, J., Studerus, E., Casalicchio, G., and Jones, Z. M.: mlr: Machine Learning in R, Journal of Machine Learning Research, 1–5 pp., 2016.

Breiman, L.: Random Forests, Mach Learn, 45, 5–32, 2001.

Deems, J. S., Painter, T. H., and Finnegan, D. C.: Lidar measurement of snow depth: A review, https://doi.org/10.3189/2013JoG12J154, July 2013.

Harder, P., Pomeroy, J. W., Helgason, W. D., and Helgason, W. D.: Improving sub-canopy snow depth mapping with unmanned aerial vehicles: Lidar versus structure-from-motion techniques, Cryosphere, 14, 1919–1935, https://doi.org/10.5194/tc-14-1919-2020, 2020.

Jacobs, J. M., Hunsaker, A. G., Sullivan, F. B., Palace, M., Burakowski, E. A., Herrick, C., and Cho, E.: Snow depth mapping with unpiloted aerial system lidar observations: A case study in Durham, New Hampshire, United States, Cryosphere, 15, 1485–1500, https://doi.org/10.5194/tc-15-1485-2021, 2021.

Probst, P., Wright, M. N., and Boulesteix, A. L.: Hyperparameters and tuning strategies for random forest, https://doi.org/10.1002/widm.1301, 1 May 2019.

Renette, C., Olvmo, M., Thorsson, S., Holmer, B., and Reese, H.: Multitemporal UAV lidar detects seasonal heave and subsidence on palsas, Cryosphere, 18, 5465–5480, https://doi.org/10.5194/tc-18-5465-2024, 2024.

Zhang, K., Chen, S. C., Whitman, D., Shyu, M. L., Yan, J., and Zhang, C.: A progressive morphological filter for removing nonground measurements from airborne LIDAR data, IEEE Transactions on Geoscience and Remote Sensing, 41, 872–882, https://doi.org/10.1109/TGRS.2003.810682, 2003.

---

## Author Comment (AC2)

**Point-by-point replies to the question and comments by Reviewer 2**

Dear Reviewer 2,

we are pleased to submit the replies to your questions and are thankful for the insightful comments and many good suggestions, as well as we are grateful for your time and effort in providing valuable feedback. We believe that addressing the issues raised by you, have now substantially improved our manuscript.

We hope our answers meet your approval. Your comments and our point-by-point responses are presented below. Please note, that we added a detailed description of a new RF modelling approach in the appendix A.

| Reviewer #2 comments | Action | Response |
|---|---|---|
| 1. The authors produce a random forest (RF) model based on lidar-derived topographic predictors and point observations of snow depth. The RF model is used to create a continuous snow depth field over the three independent palsas in Finland and Sweden. It is then evaluated against the point observations and compared to UAS-lidar-derived snow depths. Finally, the authors discuss the implications of snow depth variability on permafrost dynamics at the palsas. The manuscript has well-constructed figures, relies on a unique and interesting dataset, includes an assessment of a wide range of reasonable terrain predictors of snow depth, and the methods are on the right track. However, there are some significant concerns, including manuscript organization/framing, limited lidar validation/processing concerns, model overfitting, and generally weak analysis/discussion. These are detailed below. | Answered | We sincerely appreciate your thorough review, insightful comments, and thoughtful questions. Your valuable feedback has provided us with important guidance to improve our manuscript.
We believe that addressing the issues you have raised have significantly enhanced the clarity and overall quality of our work. Below, we present our point-by-point responses to your comments and suggestions. |
| 2. Lastly, please do not be overwhelmed by all of the comments! Addressing the major suggestions and proofreading the manuscript thoroughly should move this study much closer to publication. The specific comments are intended as suggestions/thoughts to help steer the revision process and are generally related to the below Major Suggestions/Comments. | Answered | Thank you for your encouraging words. We appreciate your detailed feedback and are confident that your suggestions have significantly enhanced the quality of our manuscript. |
| **Major Suggestions/Comments** | | |
| *Concerns with Research Objectives, Methods, and Manuscript Organization* | | |
| 3. Ln 94-99: It is my opinion that the research objectives need to be refined. The first stage of the paper should be an evaluation of lidar-based snow depth, followed by an evaluation | Changed/
Answered | In response to your comments and those of Reviewer 1, we refined the focus of our manuscript during the review process. Originally, our main objective was to analyse |

| | | |
|---|---|---|
| of the RF modeling approach. Only then should the authors discuss the potential implications of the depth patterns, and this should be a smaller part of the manuscript focused in the discussion. Since the authors did not explicitly collect data to link snow depth to changes in the active layer (or ice loss/gain), the outcomes are more based on expectations and assumptions – which may be valid, but to verify and to be a focus of the manuscript would require more data. The points described on Ln 97-99 are underdeveloped and unsupported by observations. | | snow depth patterns and their potential impact on palsas. However, based on your valuable suggestions, we now recognize the importance of a more explicit comparison between UAS-LiDAR and RF-modelled snow depth products.

We have adjusted our research objectives accordingly to highlight this comparison. However, we still believe it is important to discuss the potential impact of snow depth distribution on palsa dynamics. We agree, nevertheless, that these interpretations are based on assumptions and not on direct observed data. For clarity, we explicitly point out that these ideas need to be further validated in future studies. |
| 4. Section 4.3 needs revision. Unless the expected errors in the lidar product are further expanded upon, these are physical observations and it is standard practice to assume that these products have uncertainties errors proportional to the sensor error (e.g., ~5 cm). This can be directly evaluated from ground observations of snow depth and was to some degree. However, the errors were much larger than expected (>20 cm, Ln 298-302), raising concerns about the processing of lidar data to produce snow depth maps. This component and the framing of the analysis are significant concerns. A section early on evaluating the lidar depth products seems necessary and considering the influence of vegetation on their accuracy explicitly (for example, examining some of the outliers in Figure 8 more closely) – addressing the concern of vegetation compression should be added here and vegetation height models produced from the summer lidar point cloud | Changed/ Answered | Thank you for highlighting these concerns. In response, we have conducted an additional model run, refining the LiDAR-based snow depth estimation by removing vegetation from the initial LiDAR products to create a more accurate DTM. Additionally, we implemented hyperparameter tuning and tested different cross-validation folds to mitigate potential overfitting.

These methodological improvements are explained in detail in Appendix A. The updated results demonstrate a substantial improvement in the accuracy of the LiDAR-derived snow depth estimates. However, snow depth estimation remains less accurate for the *Thermokarst* point group, likely due to high reflectance in water-dominated areas.

Furthermore, the revised RF modelling approach now has improved results in terms of representation and generalization. To ensure clarity, we have revised the relevant sections accordingly and have introduced a new section specifically detailing the processing and validation of LiDAR data.

In the following, we refer to Appendix A. There you find all necessary information, which answer your questions. |
| 5. Comparing the lidar to an RF model trained and evaluated against <200 observations directly is not appropriate. As presented, the lidar depth analysis does not add much to the manuscript – I suggest it be redone (reprocessed data, more detailed lidar depth evaluation), and/or, the work reframed to simply build the RF model using lidar terrain and snow depth point observations, then a revised analysis on how these patterns are expected to influence the palsa stability. | Changed/ Answered | See Appendix A. |
| *Random Forest Modeling Concerns* | | |
| 6. Ln 189-191: It seems like little consideration was given to the hyperparameters, and | Changed/ Answered | See Appendix A. |

| | | |
|---|---|---|
| several important ones (like maximum split size, and minimum node size) are not mentioned. Please clearly state the hyperparameters used, and an optimization routine should be included to select these – not just using defaults – which are likely geared towards a much larger data set. If done correctly, this will reduce overfitting (see following concerns) | | |
| 7. Various model runs were not clear. The predictors for model 1,2, and 3 should be explicitly stated, with the appropriate reasoning within the methods section | Changed | We have described the input parameters used for each model run in lines 205–209. However, based on your (and Reviewer 1) suggestions and the revised model design, we have decided to focus on a single model run. As a result, we used only 12 parameters and removed the information related to previously unused parameters and former model runs. The section on input parameters and model runs has been updated accordingly. |
| 8. 10-fold cross-validation is not sufficient to ensure that the model is not overfit. Each model is still trained with 90% (9/10) of all data (and the training dataset is relatively small <200 snow depths). The authors should explore the influence of fewer folds (e.g., 3-10) to assess how much model performance is degraded. For such a small dataset, around four (4) folds seems more appropriate in this case | Changed/ Answered | See Appendix A. |
| 9. Ln 232 – 242: This is relevant to RF model training/evaluation – I suggest moving to the random forest training section | Changed | We agree and moved this part to the description of the RF algorithm and modelling data preparation. |
| 10. The results shown in Table 3 (R2 > 0.99, RMSE less than the expected measurement uncertainty <3 cm) suggest substantial overfitting of the random forest model | Changed/ Answered | See Appendix A. |
| 11. Avoid analysis based on terms like 'potentially,' 'possibly,' and 'probably' -- you should focus your study on explaining and describing the data you have collected its likely implications with clearly stated support | Changed | We agree and changed/avoided assumptions within the abstract/discussion at lines 18, 320, 323, 347. We either supported our ideas by own observations in the field (decreasing parts, block erosion) or pointing out the uncertainties and the necessity to verify these assumptions in further studies. |
| 12. The terminology of cooling and warming spots was unclear – since these are not a standard term to my knowledge, these need to be explicitly defined early on and used consistently throughout the manuscript | Changed | Thank you for highlighting this important point. To ensure clarity and consistency, we have explicitly defined these terms in the introduction when explaining the role of snow dynamics in palsa environments:

We define *Cooling Spots* as areas on and around a palsa where the snow cover remains relatively thin during winter. Due to the lack of insulating snow, these areas experience increased heat loss from the ground to the atmosphere, allowing frost to penetrate deeper into the subsurface. As a result, when |

| | | |
|---|---|---|
| | | summer arrives, the snow in these areas melts earlier, exposing the ground to warm temperatures for an extended period. This prolonged exposure leads to a deeper thaw of the active layer, making these locations more susceptible to ground cracking and thermokarst formation, particularly along the palsa edges. Cooling spots are typically found in elevated or wind-exposed areas of the palsa, where snow accumulation is naturally limited.

In contrast, *Warming Spots* are areas where a relatively thick snow cover accumulates during winter. The insulating properties of the snow reduce heat loss from the ground, preventing deep frost penetration and keeping the underlying soil comparatively warmer throughout the winter. In summer, the accumulated snow melts later, delaying the warming of the ground and slowing active layer thawing. Consequently, the active layer in these areas tends to be shallower compared to cooling spots. Warming spots are typically located in depressions, concave terrain, or wind-sheltered locations where snowdrifts form.

These definitions have been integrated into the manuscript to provide a clearer conceptual framework for our analysis. |
| 13. Ln 71-73: UAS-lidar-based snow depth monitoring approaches/literature should be sufficiently reviewed in the introduction (& by the authors). The approaches used to produce snow-depth products do not align with standard practice (e.g., classifying the vegetation-free ground surface). See work by Avanzi et al., 2018; Harder et al., 2020; Jacobs et al., 2021. | Changed | We acknowledge the importance of previous studies that have applied UAS-LiDAR for snow depth mapping and appreciate your suggestion. Our initial focus was primarily on demonstrating the feasibility of snow depth modelling using RF and assessing its implications for palsas. However, as the study evolved, the focus shifted more towards a comparative analysis between Random Forest modelling and LiDAR-based snow depth estimation. In response to your suggestion, we have incorporated additional references on UAV LiDAR-based snow depth mapping. Specifically, we adopted the vegetation removal approach inspired by Jacobs et al. (2021) and further reviewed relevant studies, including those by Avanzi et al. (2020) and Harder et al. (2020). |
| 14. Grammatical and sentence structure issues limit communication effectiveness in the paper. A thorough proofreading by a third party before resubmission would benefit the paper. Some specific instances of this were noted in the comments below | Changed | Thank you for pointing this out. We will have a close look to grammar and sentence structure before resubmission and will particular improve the comments you are mentioning below. |
| **Minor/Technical/Grammatical Suggestions** | | |

| | | | |
|---|---|---|---|
| 15. | Stick with snow depth or snow height throughout – be consistent with word choice | Changed | Agree, we are now using the term snow depth and changed snow height in lines 258, 264 and 266. |
| 16. | The use of the word 'precision' is questionable at times (see abstract Ln 9). Precision measures the ability for repeatable measurements. Accuracy is a better term for assessing something like a random forest model. Read through the manuscript and consider if the use of 'precision' is appropriate throughout | Changed | This is a good point. We changed the term precise/precision/precisely in lines 9, 74, 90, 97, 188, 288, 316, 365, 370 and 396. |
| 17. | Word choice should be reviewed throughout – (e.g., Ln 78: 'very strong changes' could be 'control', Ln 212: 'realism') | Changed | We scanned our manuscript for judgmental words and changed them to more neutral terms. |
| 18. | Avoid broad terminology throughout, especially before something a term is explicitly defined (e.g., Ln 150 – was not clear what 'input parameter data' referred to again at Ln 181, 182, 185). I suggest defining the types of input parameters earlier on – like was done in Section 3.3 | Changed | By "input parameters", we are specifically referring to the variables listed in Table 2. These parameters are all derived from the initially generated DSM in summer and were exclusively used in the RF model. To enhance clarity and avoid ambiguities, we have explicitly defined the term "input parameters" in line 144. Additionally, we have standardized the terminology by removing the word "data" in line 150 to ensure consistency throughout the manuscript. |
| *Abstract* | | | |
| 19. | Ln 13-15: Machine learning is used to model the snow depth spatially and relies on observations. On its own, it does not capture snow depth patterns. Consider rewording | Changed | We agree and changed the sentence: "This considerable difference highlights the capability of machine learning to model fine-scale snow distribution based on in-situ observations." |
| 20. | The abstract should be a single cohesive paragraph, avoid splitting into two parts | Changed | We agree and changed the abstract to a single cohesive paragraph. |
| *Introduction* | | | |
| 21. | Ln 26, 57-58: While snow cover duration is decreasing, the suggestion that snow depth is increasing substantially in these regions is less clear. This paper suggests snowfall extremes will be reduced in the study area (https://www.nature.com/articles/s41598-021-95979-4) – can you clarify this point? | Changed/ Answered | Thank you for your suggestion. We acknowledge that while snow cover duration is generally decreasing, trends in snow depth are more regionally variable. Increased winter precipitation may lead to higher snow depths in some areas, whereas other regions might experience a decrease in snowfall extremes, as suggested by Quante et al. (2021). Since we have not specifically analysed these trends for our study region, we recognize the need to be more cautious with this statement and have revised it accordingly to reflect the regional variability and associated uncertainties. |
| 22. | Ln 32-33: Sentence structure/clarity issues – please revise | Changed | We changed the sentence: *In northern Fennoscandia, particularly in northern Finnish Lapland - the main focus of this study - specific periglacial permafrost landforms known as palsas are at risk of disappearing within this century (Leppiniemi et al., 2023).* |

| | | | |
|---|---|---|---|
| 23. | The relevance of palsas is not addressed clearly in the introduction. Please add some sentences on their general significance, e.g., Do they stabilize permafrost? Provide habitat? Have societal relevance? | Changed | We have added that palsas serve as indicators of climate warming, as their degradation and disappearance reflect rising temperatures (Leppiniemi et al., 2023). Additionally, they provide important habitats for various animal species (Luoto et al., 2004) and hold significant cultural and societal relevance for the Sámi people, particularly in the context of traditional reindeer herding (Markkula et al., 2019). |
| 24. | Ln 53: remove 'exemplarily' | Changed | Removed. |
| 25. | Ln 56-57: These points seem essential for understanding the relevance of palsas – I suggest this is moved earlier in the introduction when palsas are defined | Changed | We moved the sentence "*Microtopography affects snow depth and creates an environment, in which the palsas usually receive enough penetrating cold air to remain stable and to last year after year due to a thin snow cover.*" to line 41. |
| 26. | Ln 60-61: 'in-situ measured data' or 'observations' need to be clarified. Is this temperature data? Snow depth? Other? | Changed | We are referring to "snow depth data" an inserted this term for clarification. |
| 27. | Ln 78-79: Wording is unclear – '…limits information value of satellite data…' | Changed | We changed the sentence: *Small-scale structures, such as palsas, exhibit significant variations in snow depth at fine spatial scales, which reduces the usefulness of satellite data for analysing small-scale processes in these structures.* |
| 28. | Ln 81:'Another' should start a new paragraph – this section is also very short relative to the prominent role that machine learning plays in the paper. I suggest adding more detail. | Changed | We would like to point out kindly, that there is a paragraph between line 80 and 81.
As mentioned previously, we initially focused more on the impacts of the snow depth distribution to palsas in this paper. However, after your useful comments, we agree and inserted more details about machine learning and specifically RF. |
| 29. | Ln 90: '…test methods for generating detailed snow distribution maps..'' should lead this section. The objectives need to be clearly stated up front | Changed | Yes, we agree and have adjusted the focus of the paper accordingly. Specifically, we have ensured that the section begins with a clear statement of our objectives, emphasizing the evaluation of methods for generating detailed snow distribution maps. |
| *Data and Methods* | | | |
| 30. | Ln 141: a comprehensive dataset of what? Specify | Changed | We have clarified this statement by specifying that we collected a comprehensive dataset consisting of UAS-LiDAR data and in-situ snow depth measurements for modelling purposes. |
| 31. | Figure 3: Should clearly state the actual observations that were collected | Changed | We have clarified that the collected data include UAS-LiDAR measurements, which were used to generate DTMs for both winter and summer, as well as in-situ snow depth measurements, which served as training data for the modelling. |
| 32. | Ln 151: This is the first time LiDAR is mentioned. Needs to be introduced within the introduction | Changed | We agree and introduced LiDAR in lines 73 – 74 in the context of the studies by Rauhala et al. (2023) and Meriö et al. (2023). |

| | | |
|---|---|---|
| 33. Ln 157-160: very confusing. No SfM, but then orthophotos were created? That relies on photogrammetry -- but then you state point cloud densities. Are these associated with lidar or RGB orthophotos? If lidar, need to put it right after the lidar. Also, should report density per square meter as it is the standard. | Changed | We acknowledge the potential for misunderstanding and have clarified our statement in lines 157/158. Specifically, we used SfM techniques solely for the creation of orthophotos. The RGB flights were conducted using an Autel EVO II Pro V2 UAV at a flight altitude of 80 m, with a 75% overlap for each flight. Initially, we had stated that the orthophotos were acquired with the integrated RGB sensor of the LiDAR mapper. However, this was a misunderstanding, and we have now corrected this statement. The orthophotos do not contribute specific data to the analysis but were solely used for figure creation. |
| 34. Ln 162-164: Revise sentence structure for clarity | Changed | We changed the sentence: "*By substracting the winter by the summer DSM in Geographic Information Systems (GIS) – ArcGIS Pro by Esri was used – snow depth distribution datasets were calculated, allowing the comparison of UAS-LiDAR snow depth (SD$_{LiDAR}$) and RF modelled (SD$_{RF}$).*" |
| 35. Ln 166: should be 'by an RTK GPS system' | Changed | We agree and added the term. |
| 36. Ln 170: word choice - 'optimal' | Changed | We changed "optimal" to "diverse". |
| 37. Ln 172: The sampling strategy is claimed to be randomized, though it appears observations were collected along transects with some random points. Some of these could be biased, so it would be useful to add a bit more description. There are also areas with clear gaps | Answered | We acknowledge that the sampling strategy may appear structured, potentially suggesting a bias. However, no strict transect approach was followed when measuring snow depth in Pousu. Instead, the sampling locations were selected based on terrain features, as illustrated in the figure in Appendix E, which we will include in the manuscript appendix. Measuring snow depth under these environmental conditions is challenging, and our data collection was constrained by a limited time frame. Therefore, we prioritized a well-distributed dataset that captures the variability within our palsa sites as effectively as possible. |
| 38. Related to the previous point, the distribution of snow depth observations included in the appendix should be split by site (in my opinion) | Changed | See Appendix D. |
| 39. Ln 176: It isn't easy to make out any snow-free areas on the palsas in the imagery – can these be indicated? | Answered | The snow-free points represent extreme locations in highly exposed areas. These points were specifically captured to ensure that the model is trained with the full range of snow depth variations observed in the field. However, due to the limited resolution of the orthophotos, it is difficult to clearly visualize these areas as the images appear too blurry to clearly highlight them. However, these snow-free areas are mainly located on steep slopes where wind-induced redistribution of snow and downslope movement have either |

| | | | removed or significantly reduced the snow cover. |
|---|---|---|---|
| 40. Ln 184-188: To clarify, the full training set is based on only 185 observations – but increased due to the buffering? Please indicate how many unique features were actually used to train the model after the buffering. Will help the reader understand the robustness of the model | Changed/ Answered | See Appendix A. | |
| 41. Ln 193-194: Just state the metrics were normalized 0-1, with the highest output importance set as 1 | Changed/ Answered | See Appendix A. | |
| 42. Ln 205: The removal of elevation as a predictor needs more explanation – the logic that is will 'reduce possible overfitting' is not apparent | Changed/ Answered | Please refer to our previous responses, especially comment #7, regarding the removal of the initial model runs. As a result, it is no longer necessary to elaborate on the exclusion and reintroduction of the *Elevation* parameter. For clarity regarding our initial approach: *Elevation* was excluded in the second model run because all other input parameters were derived from it. This step was taken to assess whether *Elevation* might introduce bias into the modelling results. After analysing the outcomes, we found no indication of such bias and subsequently decided to retain *Elevation* as an input parameter in the final model. | |
| 43. Ln 206-207: wordy, what is 'initial minimal impact'? | Changed | We changed the term to "low impact". | |
| 44. Ln 211: Unclear how this offers a balanced representation. The idea of taking an area is usually to remove noise, reduce the influence of sampling or geolocation errors, and to grow the training set size (taking groupings of nearby points vs. a single one - which should improve the robustness of the model). Please explain further. | Changed/ Answered | Thank you for your comment. We agree that the phrase "balanced representation" was not the most precise wording. To clarify, the buffering strategy was implemented to reduce noise, minimize the influence of geolocation and sampling errors, and enhance the robustness of the model by increasing the number of training points. By incorporating groupings of nearby points rather than relying on single-point measurements, this approach helps improve the model's stability and realism, as demonstrated in Bergamo et al. (2023). We have revised the manuscript accordingly to reflect this explanation more clearly. | |
| 45. Table 2: Nice table! For features like TPI (which are determined to be very important), you should be more detailed in their definition. More than 'it combines several topographic features.' TPI is generally just the relative elevation of a point to surrounding points within some radius (or adjacent pixels) | Changed | Thank you for your positive feedback! We agree with your suggestion and have added more detailed explanations for *TPI, Wind Effect, Valley Depth, Channel Network Base Level,* and *Wind Exposition* to ensure clarity and precision in their definitions. | |
| 46. Ln 238-239: Be careful with wording. Correlation (strength of linear relationship) and significance (based on statistical testing) are not the same thing | Changed | We recognize the difference between correlation and statistical significance and have adjusted the wording accordingly to ensure accuracy. In particular, we clarify that | |

| | | | the analysis aimed to assess the strength of the relationships between the input parameters and the SD$_{RF}$ predictions, rather than implying statistical significance unless explicitly tested. |
|---|---|---|---|
| *Results* | | | |
| 47. | Section 4.1: Nice job describing results clearly and sequentially by site. | | Thank you very much! |
| 48. | Figure 5: Nice figure, it would be useful to add annotations for areas of interest referred to in Section 4.1 on the figure (e.g., the collapsed areas) | Answered | Thank you for your valuable suggestion! While we acknowledge that adding annotations could enhance interpretability, we aim to maintain clarity and avoid overloading the maps with excessive information. Additionally, we want to prevent cross-referencing multiple figures within a single visualization.
 For these reasons, we have decided not to modify Figure 5 further but will ensure that the areas of interest are clearly described and referenced within the text. |
| 49. | Ln 252: When stating things like 'slightly higher,' specify the magnitude (is this 10cm, 20cm, 5cm?). Same as Ln 257, how much lower? | Changed | We agree with your suggestion and have included exact numerical differences in centimetres to provide a more precise comparison in lines 252 and 257. |
| 50. | Ln 272-273: sentence clarity issue | Changed | We changed the sentence: Notably, deviations in the areas surrounding the palsas are primarily characterized by higher snow depths predicted by the RF model. |
| 51. | Figure 7: Nice figure! Be sure to add more specifications on the model runs in the methods section | Answered | See comment #42. |
| 52. | Ln 294: How were they separated into 'point groups' used to produce Table 4 – how were the different areas delineated and can these be added to the maps? | Changed/ Answered | See Appendix A and E. |
| 53. | Ln 310-313: Correlation analysis results should be included as a table – this could be added to the appendix if the authors do not want to include it in the body of the paper | Changed/ Answered | See Appendix F. |
| *Discussion* | | | |
| 54. | Ln 318-319: Revise based on previous comments | Answered | Revision have been done. |
| 55. | Ln 329: warming and cooling spots need to be defined more before this point. What is a good technical definition? For example, are warming spots where the net heat flux into the ground during the winter is highest - making these areas warmer? vs. Cooling spots, where the net heat flux into the ground is lowest? We need to have a clear and more scientific definition | Changed/ Answered | See answer to comment #12. |
| 56. | Ln 345: 'Cooling spots inhibit a greater active layer thickness in summer' – is this the technical definition? It comes across as difficult to interpret. An alternative version of this: 'Cooling spots result in shallower active | Changed/ Answered | We appreciate your suggestion and have revised the sentence accordingly to improve clarity: |

| | | |
|---|---|---|
| layers in summer compared to warming spots.' | | "Cooling spots result in shallower active layers in summer compared to warming spots." Additionally, we have provided a detailed definition of cooling and warming spots in response to comment #12 to ensure consistency throughout the manuscript. |
| 57. Figure 8 - Nice figure. The delineations are helpful. Similar delineations would help the interpretation of results in prior figures | Answered | See answer to comment #48. |
| 58. Section 5.2: This section should be revised thoroughly – see previous comments on lidar snow depth and RF model comparison | Answered | We have revised this section based on our new results. |
| 59. Ln 354: Luo and Panda studies were based on satellite remotely sensed snow cover – not sure I understand the link to UAS-lidar observations. Also, not clear what 'not in depth post-processed data' is. I did not understand the transition of the discussion from snow depth to snow cover | Changed | Thank you for pointing this out. We acknowledge that the studies by Luo et al. (2022) and Panda et al. (2022) focus on satellite-based snow cover observations rather than snow depth. To avoid confusion, we have removed these references in this context and ensured that our discussion remains focused on snow depth mapping. Additionally, since our revised model approach now utilizes a DTM instead of a DSM, we have removed the statement regarding "not in-depth post-processed remote sensing data" to accurately reflect the improved data processing methodology. |
| 60. Ln 363-364: How did manual probing address the issue of vegetation? The uncertainty in these observations was never discussed | Answered | Thank you for highlighting this important point. We acknowledge that the impact of vegetation on manual snow depth probing was not explicitly discussed. In our study, manual probing was conducted with a heavy yardstick, which allowed us to reach the ground despite the presence of vegetation. However, we recognize that vegetation, particularly tall grasses and shrubs, can introduce uncertainties in snow depth measurements. In areas with denser vegetation, there is a possibility that the probe may not always reach the exact ground surface, leading to slight overestimations of snow depth. To address this, we have expanded our discussion on potential uncertainties in manual snow depth measurements and their implications for model accuracy. |
| 61. Ln 366-368: This doesn't seem like only a lidar limitation - but a measurement challenge in general. Measuring snow over dense vegetation with air voids, compression, etc.. is always challenging. New approaches to correct the lidar based on the underlying vegetation type/density/height may improve lidar snow depth products. | Answered | We agree that the challenges of measuring snow depth over dense vegetation are not solely a limitation of LiDAR but rather a general measurement issue. We acknowledge that new approaches, such as correcting LiDAR-based snow depth estimates based on vegetation type, density, and height, could improve the accuracy of these products. We briefly addressed this in the discussion and highlight it as a potential avenue for future research. |

| 62. | Ln 375-387: The discussion in this paragraph was strong, and it was easier to follow the logic. This could be an example to use when revising the discussion. | Answered | Thank you! We have now used this paragraph as example for revising the discussion. |
|---|---|---|---|
| 63. | Ln 392, 407-408: Why was vegetation not removed from the summer point cloud? I do not understand why this was done in this manner. This step is critical for snow depth mapping with lidar. | Changed/ Answered | See Appendix A. |
| 64. | Ln 395-397: There is a growing body of literature on this that would be useful to review. See Buhler 2016, 2017; Adams et al., 2018; Avanzi et al., 2018, Cho et al., 2024 (Preprint), Eker et al. 2019; Harder et al. 2020 (compares lidar and RGB) | Answered | We have reviewed the recommended literature and incorporate relevant findings or ideas if they make a meaningful contribution to the context of our study. |
| 65. | Much of the discussion relies on findings from other studies and assumed links to snow depth observed in this study to conclude – not clear to me what value the work presented here has to understanding palsa permafrost dynamics more than point observations on a transect across one of these features would. Related to previous comments on reframing and refocusing the research objectives | Answered | Thank you for your comment. We recognize that directly linking our modelled snow distribution to permafrost dynamics remains a complex challenge. However, we believe that our study offers significant value beyond point transect measurements by providing the first spatially continuous snow depth maps over palsas using validated LiDAR and RF-based approaches. Our results show that these models perform well compared to independent validation datasets, confirming the reliability of the derived snow depth distributions. Given the crucial role of snow in regulating permafrost stability, we argue that these spatial datasets provide valuable insights into the potential snow-induced thermal dynamics of palsas. While additional ground-based validation of permafrost responses would strengthen this link, our study provides an important foundation for future research in this area. We have clarified these points in the discussion to better emphasize the unique contribution of our study. |
| 66. | Ln 412-414: A fewer number of folds should be used in the model training/validation | Changed/ Answered | See Appendix A. |
| 67. | Ln 421, 425-426: A large number of input features are used in this model and the results as presented show nearly perfect model performance – are you suggesting others should be included? If others could make the model better, why were they not included? | Answered | Thank you for your comment. With our revised model approach, we now use only 12 input parameters, ensuring a more streamlined and interpretable model. Our intention was not to suggest that additional parameters should necessarily be included in this study, but rather to acknowledge that future research could explore further potentially relevant predictors. For instance, more detailed vegetation classifications - such as specific vegetation types or density indices - could enhance snow depth modelling. Additionally, there may be other influential parameters that are not directly linked to snow depth but still play a role in snow distribution patterns. |

| | | Identifying such factors would require a dedicated study focused on assessing and selecting the most critical parameters for snow depth modelling. |
| | | We have clarified this point in the discussion to ensure that our statement is not misinterpreted as a recommendation for additional parameters in the current model. |
| 68. Ln 423-424: Good point | | Thank you! |
| 69. Once noted challenges throughout are addressed – the discussion should be re-written to align with the updated manuscript | Answered | After all changes have been made, we have adapted the discussion based on the updated manuscript. |
| *Conclusions* | | |
| 70. As presented, the paper is focused on the evaluation of the methods for snow depth mapping and on the predictors that control the depth distribution -- discussion into the influence of these characteristics on the thermal profiles is purely assumption based -- thus reframing the conclusion in line with the revised paper and the actual results/data presented will be critical in the revised version. | Answered | Based on all the revisions and refinements made throughout the manuscript, we have rewritten the conclusion to align more clearly with the revised focus of the paper and the actual results presented. This ensures that our conclusions remain grounded in the data and analyses conducted. |

**Appendix**

In this section, we provide additional information addressing comments #4, 5, 10, 16, 22, 23, 24, 44, 49, 50, 54, 56 from Reviewer 1 and #5, 6, 8, 10, 38, 40, 41, 52, 53, 63, 66 from Reviewer 2.

We sincerely appreciate your insightful comments and suggestions, which have significantly contributed to improving both the modelling approach and the overall quality of the manuscript.

**Appendix A**

To ensure high-quality modelling results and accurate snow depth distribution maps derived from UAS-LiDAR, we implemented your recommendations, including the removal of vegetation from the LiDAR-derived products and a re-evaluation of the modelling approach.

Additionally, we incorporated hyperparameter tuning and cross-validation to determine the most suitable parameter settings for the Random Forest model. To further improve model robustness and prevent overfitting, we also adjusted the data splitting strategy by testing the RF model on an independent external dataset.

1. Removal of vegetation from UAS-LiDAR DSM

Our initial decision to retain vegetation in the modelling process assumed that small and dense vegetation, as present in our study sites, is difficult to remove - even from point clouds. Testing several vegetation filter

algorithms, such as the *Cloth Simulation Filter* (CSF) and *Statistical Outlier Removal* (SOR) in CloudCompare, confirmed this assumption, as the vegetation was not properly removed in the resulting products.

Additionally, we considered that vegetation significantly influences snow depth distribution by enhancing snow retention capacity. Therefore, we initially decided to include vegetation in the modelling process, expecting it to be beneficial for RF modelling.

However, based on your suggestions, we tested the *Progressive Morphological Filter* (PMF) Algorithm as described by Zhang et al. (2003) and Jacobs et al. (2021) and obtained satisfactory results with an effective removal of vegetation. We applied PMF filtering using the following parameters:

- Window sizes: 0.5, 1, 2, and 3
- Thresholds: 0.05, 0.1, 0.3, and 0.5

The extracted ground and vegetation points were saved in point cloud format. Using CloudCompare, we generated a DTM for each palsa using the Rasterize function. Empty cells within the point clouds were interpolated with a triangle max edge length value of 5.0.

The newly created DTMs were then used to recalculate the snow depth distribution for all three test sites in GIS, following the methodology described in the manuscript. In our initial calculations, all negative values were set to zero. However, in this revised approach, we retained negative values to highlight areas where either the LiDAR sensor produced inaccuracies or surface degradation occurred between the summer and winter flights.

Based on these refined DTMs, we recalculated all input parameters used in the final RF model run in SAGA GIS. The following 12 parameters were included: *Aspect, Elevation, Channel Network Base Level, Channel Network Distance, Negative Openness, Positive Openness, Relative Slope Position, Slope, Topographic Position Index, Valley Depth, Wind Effect, Wind Exposition*.
A detailed description of these parameters is provided in Table 2. We have now focused on a single model run, and accordingly, we have removed descriptions of other parameters from the manuscript to ensure clarity and consistency.

2. Splitting data into training and test datasets

In the initial study design, we used the entire buffered $SD_{in-situ}$ dataset to extract the input parameters from the raster stack, resulting in a data frame with 5222 points. We then split this dataset into 70% training and 30% test data. However, this approach introduced a risk of overfitting, as each $SD_{in-situ}$ point was represented an average of 28 times in the dataset. Consequently, many points appeared in both the training and test datasets, reducing the independence of the validation process.

To address this issue, we revised our study design by first separating 70% of the point features from each $SD_{in-situ}$ dataset for training and 30% for testing. Only after this separation did we extract the input parameter values for the training dataset, ensuring a clear distinction between training and validation data. The test dataset was reserved exclusively for model validation. The following extract from the R script illustrates these steps:

```
**===== Function to split training and test dataset =====**
split_shapefile <- function(shp) {
  set.seed(42)  # Ensure reproducibility
  num_samples <- nrow(shp)  # Get the number of samples
  train_indices <- sample(num_samples, size = round(0.7 * num_samples))  # Select 70% of the samples for training
  test_indices <- setdiff(1:num_samples, train_indices)  # The remaining 30% for testing

  shp_train <- shp[train_indices, ]  # Create training dataset
  shp_test <- shp[test_indices, ]  # Create test dataset

  return(list(train = shp_train, test = shp_test))  # Return the split datasets as a list
}

**Splitting the dataset for all three locations**
split_pousu <- split_shapefile(shp_pousu)
split_peera <- split_shapefile(shp_peera)
split_puolikkoniva <- split_shapefile(shp_puolikkoniva)

**Combine training and test datasets for all Palsas**
shp_train_all <- rbind(split_pousu$train, split_peera$train, split_puolikkoniva$train)  # Merge training datasets
shp_test_all <- rbind(split_pousu$test, split_peera$test, split_puolikkoniva$test)  # Merge test datasets
```

After extracting the input parameters from the raster stack, the final dataset consisted of:

- Training dataset: 3,645 points (Puolikkoniva: 1,983; Pousu: 905; Peera: 757)
- Test dataset: 1,577 points (Puolikkoniva: 836; Pousu: 401; Peera: 340)

To prevent errors and miscalculations, all NoData values were removed from the datasets, resulting in a final training dataset of 3,504 points and a final test dataset of 1,548 points for further modelling and validation.

3. Hyperparameter tuning and cross validation

To determine the optimal values for *mtry*, *min.node.size*, and *sample fraction*, we performed hyperparameter tuning using the *mlr* package in R (Bischl et al., 2016).

To prevent overfitting, we restricted the search range for *min.node.size* to 10–15 and for *sample fraction* to 0.7–0.85, following the recommendations of Probst et al. (2019) and Breiman (2001). Allowing an unlimited search range initially resulted in better model performance, but at the cost of reduced generalization, indicating signs of overfitting. We selected the final search range based on multiple test runs with different settings.

For cross-validation, we tested different fold sizes to identify the most effective configuration. The best results were achieved using a 4-fold cross-validation. The following R script extract provides details on the tuning process:

```
**===== Hyperparameter Tuning with tuneRanger (Regression) =====**

**Define the regression task**
task <- makeRegrTask(data = all_train, target = "Class")

**Define the cross-validation strategy**
cv_desc <- makeResampleDesc("CV", iters = 4)  # 4-fold cross-validation

**Define the Random Forest learner with hyperparameters as tuning options**
learner <- makeLearner("regr.ranger", num.trees = 1000)

**Define the hyperparameter search space**
param_set <- makeParamSet(
  makeIntegerParam("mtry", lower = 2, upper = ncol(all_train) - 1),   # Number of variables to consider at each split
  makeIntegerParam("min.node.size", lower = 10, upper = 15),   # Minimum number of observations per node
  makeNumericParam("sample.fraction", lower = 0.7, upper = 0.85)   # Proportion of samples used in each tree
)

**Define the tuning control (e.g., Bayesian optimization or random search)**
control <- makeTuneControlRandom(maxit = 70)   # 70 iterations for tuning

**Hyperparameter tuning with cross-validation**
tuned_params <- tuneParams(
  learner = learner,
  task = task,
  resampling = cv_desc,   # 4-fold CV
  par.set = param_set,
  control = control,
  measures = rmse   # Root Mean Squared Error as the performance metric
)

**Display results**
print(tuned_params)

**Best Random Forest model with tuned parameters**
best_learner <- setHyperPars(learner, par.vals = tuned_params$x)
```

The final tuned hyperparameters were as follows:

- mtry: 9
- min.node.size: 10
- sample fraction: 0.79

4. Permutation Importance (PI)

In our initial study design, we conducted the RF modelling once and directly used the permutation importance (PI) values provided by the model.

In our revised approach, we refined this process by repeating the calculation 100 times to obtain a mean PI value for each input parameter, ensuring more robust and reliable importance rankings.

The following R script extract details the implementation of this approach:

```
**========== Permutation Importance**
num_repeats <- 100
importance_values <- matrix(NA, nrow = num_repeats, ncol = ncol(all_train) - 1)

for (i in 1:num_repeats) {
  cat("Iteration:", i, "\n")

  # Train the model using the identical hyperparameters from tuning
  temp_model <- ranger(
    x = all_train[, -ncol(all_train)],
    y = all_train$Class,
    mtry = tuned_params$x$mtry,   # Optimized mtry value
    min.node.size = tuned_params$x$min.node.size,   # Optimized min.node.size
    sample.fraction = tuned_params$x$sample.fraction,   # Optimized sample.fraction
    num.trees = 1000,
    importance = "permutation",
    seed = i   # Different seed per run for robustness
  )

  # Store the feature importances in the matrix
  importance_values[i, ] <- importance(temp_model)
}

**Compute the mean Permutation Importance over the 100 runs**
mean_importance <- colMeans(importance_values)
```

We modified Figure 7 to display only the 12 selected parameters along with their respective mean PI values over 100 iterations. Additionally, we normalized the values, setting the most important parameter (Topographic Position Index) to 1.

[Figure]

*Figure 7. Overview of normalized mean Permutation Importance values from RF modelling over 100 iterations.*

5.   Final results and validation

Both the RF-based and UAS-LiDAR-based results were validated using the initially separated test dataset. Additionally, we conducted three further RF model runs, where in each iteration, two palsa sites were used as the training dataset, and one was used as the test dataset. This approach further validated the generalization capability of the model.

The validation results indicate that the RF-based approach now exhibits lower peak accuracies compared to the initial study design. However, by reducing overfitting, the results are more plausible and robust, while still achieving high accuracy and outperforming the UAS-LiDAR-based approach:

Table 3. Overview of the calculated Root Mean Square Error (RMSE), Coefficient of Determination (R2), Mean Absolute Error (MAE) and Standard Deviation (SD) for RF- and UAS-LiDAR-based snow depth estimations. Additionally, external validation results (RMSE and $R^2$) for RF-based snow depth at each palsa site (Peera RF, Pousu RF, Puolikkoniva RF) are provided.

| Parameter | RF | LiDAR UAS | Peera RF | Pousu RF | Puolikkoniva RF |
|---|---|---|---|---|---|
| RMSE | 18.33 | 23.49 | 16.67 | 21.31 | 27.13 |
| $R^2$ | 0.77 | 0.691 | 0.628 | 0.767 | 0.578 |
| MAE | 13.26 | 17.49 | - | - | - |
| SD | 18.11 | 20.84 | - | - | - |

We recalculated all metrics for different point groups and included the number of points per group. These groups were classified visually, based on orthophotos, slope data, and elevation characteristics of the respective locations.

The results show that the accuracy differences between RF and UAS-LiDAR-based approaches are now less pronounced. However, in certain categories, such as *Thermokarst* and *Open Area*, the UAS-LiDAR-based results show lower accuracy, likely due to measurement inaccuracies caused by water surfaces and irregularities in areas with higher vegetation.

Table 4. Overview of RMSE, $R^2$, MAE and SD divided by validation point locations within the investigation areas.

| | RMSE | | $R^2$ | | MAE | | SD | |
|---|---|---|---|---|---|---|---|---|
| | RF | LiDAR | RF | LiDAR | RF | LiDAR | RF | LiDAR |
| On Top (n = 69) | 8.33 | 8.33 | 0.841 | 0.730 | 3.84 | 3.84 | 8.32 | 10.83 |
| Edge (n = 66) | 13.12 | 13.12 | 0.894 | 0.768 | 5.85 | 5.85 | 12.82 | 19.09 |
| Thermokarst (n = 16) | 10.99 | 33.73 | 0.893 | 0.592 | 5.42 | 30.35 | 10.69 | 25.08 |
| Open Area (n = 26) | 4.54 | 14.23 | 0.926 | 0.519 | 1.56 | 9.84 | 4.40 | 12.59 |

Figures 5, 6, 8, and 9 have been updated based on the new results.

Figure 5 now includes the recalculated snow depth maps. We have incorporated all areas where $SD_{LiDAR}$ values are below 0, visualizing these parts in red to highlight regions where the LiDAR sensor may have measured incorrectly or where degradation has occurred between flights.

[Figure]

*Figure 5. Snow depth predictions based on the RF model (left) and the UAS-LiDAR (right) at site Puolikkoniva (a, b), Pousu (c, d) and Peera (e, f) palsas. Red points are showing the in-situ snow depth measurement locations.*

In Figure 6 we inserted the new calculated difference maps and we also included the parts with negative values in red:

[Figure]

*Figure 6. Snow depth differences between modelled and UAS LiDAR results at a) Puolikkoniva, b) Pousu and c) Peera palsas.*

Figure 8 shows the scatter plots based on the 30% test dataset. Here we used only the single values of the SD$_{in-situ}$, not considering the values within the buffer areas of the test data. We decided to do it like that, to obtain a very fine validation of both methods:

[Figure]

*Figure 8. Scatter plots with regression lines for UAS-LiDAR-derived and RF-modelled snow depths, based on the external test dataset.*

Figure 9 has been updated to reflect the new results. Additionally, we have incorporated the calculated slope derived from the DTM of Pousu palsa.

[Figure]

*Figure 9. Explanation of differences between UAS LiDAR-derived and RF-modelled snow depths.*

**Appendix B**

[Figure]

*Figure 2. Climate chart of Kilpisjärvi (FMI, 2022). Dotted line shows 2 m above ground temperature in °C, dashed line shows precipitation in mm and solid line shows snow depth in cm.*

**Appendix C**

[Figure]

*Figure 4. Snow depth measuring points within the investigation sites at Puolikkoniva (a), Pousu (b) and Peera (c) palsa illustrating different methods for recording snow depth (transects, randomized, crossed).*

**Appendix D**

[Figure]

*Figure A1. Histogram of SD$_{in\text{-}situ}$ points and respective statistics per palsa site.*

**Appendix E**

[Figure]

*Figure A2. Overview of classification of all SD$_{in\text{-}situ}$ points into classes Edge, On Top, Open Area and Thermokarst.*

**Appendix F**

*Table A3. Correlation between each input parameter and RF-modelled snow depth.*

| Parameter | Correlation to SD$_{RF}$ | Parameter | Correlation to SD$_{RF}$ |
|---|---|---|---|
| Aspect | 0.09 | Relative Slope Position | -0.49 |
| Elevation | -0.12 | Slope | 0.08 |
| Channel Network Base Level | -0.09 | Topographic Position Index | -0.87 |
| Channel Network Distance | -0.45 | Valley Depth | 0.50 |
| Negative Openness | 0.22 | Wind Effect | -0.55 |
| Positive Openness | -0.50 | Wind Exposition | -0.80 |

**References**

Avanzi, F., Zheng, Z., Coogan, A., Rice, R., Akella, R., & Conklin, M. H. (2020). Gap-filling snow-depth time-series with Kalman Filtering-Smoothing and Expectation Maximization: Proof of concept using spatially dense wireless-sensor-network data. *Cold Regions Science and Technology*, *175*. https://doi.org/10.1016/j.coldregions.2020.103066

Bergamo, T. F., Sampaio de Lima, R., Kull, T., Ward, R. D., Sepp, K., & Villoslada, M. (2023). From UAV to PlanetScope: Upscaling fractional cover of an invasive species Rosa rugosa. *Journal of Environmental Management*, *336*. https://doi.org/10.1016/j.jenvman.2023.117693

Bischl, B., Lang, M., Kotthoff, L., Schiffner, J., Richter, J., Studerus, E., Casalicchio, G., & Jones, Z. M. (2016). mlr: Machine Learning in R. In *Journal of Machine Learning Research* (Vol. 17). https://github.com/mlr-org/mlr

Breiman, L. (2001). Random Forests. *Machine Learning*, *45*, 5–32.

Harder, P., Pomeroy, J. W., Helgason, W. D., & Helgason, W. D. (2020). Improving sub-canopy snow depth mapping with unmanned aerial vehicles: Lidar versus structure-from-motion techniques. *Cryosphere*, *14*(6), 1919–1935. https://doi.org/10.5194/tc-14-1919-2020

Jacobs, J. M., Hunsaker, A. G., Sullivan, F. B., Palace, M., Burakowski, E. A., Herrick, C., & Cho, E. (2021). Snow depth mapping with unpiloted aerial system lidar observations: A case study in Durham, New Hampshire, United States. *Cryosphere*, *15*(3), 1485–1500. https://doi.org/10.5194/tc-15-1485-2021

Leppiniemi, O., Karjalainen, O., Aalto, J., Luoto, M., & Hjort, J. (2023). Environmental spaces for palsas and peat plateaus are disappearing at a circumpolar scale. *The Cryosphere*, *17*(8), 3157–3176. https://doi.org/10.5194/tc-17-3157-2023

Luo, J., Dong, C., Lin, K., Chen, X., Zhao, L., & Menzel, L. (2022). Mapping snow cover in forests using optical remote sensing, machine learning and time-lapse photography. *Remote Sensing of Environment*, *275*, 113017. https://doi.org/10.1016/J.RSE.2022.113017

Luoto, M., Heikkinen, R. K., & Carter, T. R. (2004). Loss of palsa mires in Europe and biological consequences. In *Environmental Conservation* (Vol. 31, Issue 1, pp. 30–37). https://doi.org/10.1017/S0376892904001018

Markkula, I., Turunen, M., & Rasmus, S. (2019). A review of climate change impacts on the ecosystem services in the Saami Homeland in Finland. In *Science of the Total Environment* (Vol. 692, pp. 1070–1085). Elsevier B.V. https://doi.org/10.1016/j.scitotenv.2019.07.272

Meriö, L. J., Rauhala, A., Ala-Aho, P., Kuzmin, A., Korpelainen, P., Kumpula, T., Kløve, B., & Marttila, H. (2023). Measuring the spatiotemporal variability in snow depth in subarctic environments using UASs - Part 2: Snow processes and snow-canopy interactions. *The Cryosphere*, *17*(10), 4363–4380. https://doi.org/10.5194/tc-17-4363-2023

Panda, S., Anilkumar, R., Balabantaray, B. K., Chutia, D., & Bharti, R. (2022). Machine Learning-Driven Snow Cover Mapping Techniques using Google Earth Engine. *INDICON 2022 - 2022 IEEE 19th India Council International Conference*. https://doi.org/10.1109/INDICON56171.2022.10040153

Probst, P., Wright, M. N., & Boulesteix, A. L. (2019). Hyperparameters and tuning strategies for random forest. In *Wiley Interdisciplinary Reviews: Data Mining and Knowledge Discovery* (Vol. 9, Issue 3). Wiley-Blackwell. https://doi.org/10.1002/widm.1301

Quante, L., Willner, S. N., Middelanis, R., & Levermann, A. (2021). Regions of intensification of extreme snowfall under future warming. *Scientific Reports*, *11*(1). https://doi.org/10.1038/s41598-021-95979-4

Rauhala, A., Meriö, L. J., Kuzmin, A., Korpelainen, P., Ala-Aho, P., Kumpula, T., Kløve, B., & Marttila, H. (2023). Measuring the spatiotemporal variability in snow depth in subarctic environments using UASs - Part 1: Measurements, processing, and accuracy assessment. *The Cryosphere*, *17*(10), 4343–4362. https://doi.org/10.5194/tc-17-4343-2023

Zhang, K., Chen, S. C., Whitman, D., Shyu, M. L., Yan, J., & Zhang, C. (2003). A progressive morphological filter for removing nonground measurements from airborne LIDAR data. *IEEE Transactions on Geoscience and Remote Sensing*, *41*(4 PART I), 872–882. https://doi.org/10.1109/TGRS.2003.810682

---

## Referee Report (RR1)

**Summary:**

"Comparing High-Resolution Snow Mapping Approaches in Palsa Mires: UAS LiDAR vs. Modeling" by Störmer et al. evaluates snow distribution retrieval methods in palsa mires using UAS lidar and random forest (RF) modeling across three study sites in northwestern Finland. The study highlights the role of palsas as indicators of climate change and reviews remote sensing and machine learning approaches for estimating snow depth. Both methods produced comparable results, although RF modeling showed higher accuracies over thermokarst ponds and open areas. Overall, the paper offers valuable insights into snow cover dynamics over palsas, though minor revisions could enhance its clarity and impact.

**Recommendations:**

Line 6: Change "Digital Surface Model" to "Digital Terrain Model"

Lines 15-18: It may be preferable to present the UAS lidar results first and then note the enhancements made by using RF modeling. Differentiating the two could be misleading given that the RF model relies on input parameters derived from UAS lidar data.

Figure 3: Perhaps this figure could have the first row represent the UAS lidar process and the second row represent the RF process? Or maybe specify the input parameters to make it clearer?

Figure 5: The negative snow depth from UAS-Lidar is mentioned in the Discussion, but I would suggest briefly making note of it in the Results as well. Also, changing the color of the snow depth points to indicate snow depth measurements could also be useful to directly compare to RF and lidar data.

Table 3: Can you clarify why MAE and SD are missing for each of the 3 study sites?

Line 464: It could be helpful to quantify this tighter spread, since the numbers in the previous tables seem to indicate that RF would have lower variance.

Line 555: Clarify "open water" considering it's still snow covered in the winter dataset. Noting that the lidar is scanning open water in the summer dataset could be helpful to the reader.

Line 558: Volumetric scattering does occur when lidar scans a snow surface, but the error is on the scale of <4 cm at this wavelength. I would suggest the difference in snow depth (~30 cm) is most likely from scanning the open water during the summer as more absorption is occurring over this surface.

Lines 672-673: Perhaps it could be helpful to note the snow conditions at the time of winter data collection, which would also impact scattering and lidar returns (i.e., general age of snow/grain size, presence of light-absorbing impurities, etc.).

Lines 678-682: Could you specify which wavelengths would be used to improve snow depth mapping? Visible wavelengths would penetrate deeper into the snowpack, and shortwave infrared has a higher likelihood of absorption (Deems et al., 2013).

**Minor Suggestions:**

Line 14: "...0.77 and 0.691, respectively."

Lines 130-148: Looks like there's use of both past and present tense, maybe double check for consistency.

Line 147: "...approach provides the most reliable results, and (2) how do the snow depth patterns..."

Figure 1: The font in 1b could be larger. The "Road" label could also be replaced with "Route E8" instead.

Line 160: Instead of "in the west" perhaps "extends to the west"?

Line 249: May want to double-check the grammar on this line ("-,").

Line 375: I'd recommend using either "Especially" or "directly".

Figure 6: Perhaps it could be useful to clip the results to the palsa boundaries (or at least a palsa buffer) rather than the raster extent.

Line 490: "indicate" rather than "indicating"; RF may improve the lidar results, but I would be hesitant to present the results as distinct.

Figure 9: Maybe "demonstration" rather than "explanation"?

Line 732: Similar to the previous comments, perhaps "alternative" could be rephrased as this suggests that it's an independent method.

Figure A2: I would suggest differentiating the colors more, especially the white and yellow points.

---

## Author Response (AR2)

**Point-by-point replies to the question and comments by Reviewer 2**

Dear Reviewer 2,

we are pleased to submit the replies to your questions and are thankful for the insightful comments and many good suggestions, as well as we are grateful for your time and effort in providing valuable feedback. We believe that addressing the issues raised by you, have now substantially improved our manuscript.

We hope our answers meet your approval. Your comments and our point-by-point responses are presented below. Please note that the updated figures are presented in the appendix.

| Reviewer #2 comments | Action | Response |
|---|---|---|
| **Summary:** | | |
| 1. "Comparing High-Resolution Snow Mapping Approaches in Palsa Mires: UAS LiDAR vs. Modeling" by Störmer et al. evaluates snow distribution retrieval methods in palsa mires using UAS lidar and random forest (RF) modeling across three study sites in northwestern Finland. The study highlights the role of palsas as indicators of climate change and reviews remote sensing and machine learning approaches for estimating snow depth. Both methods produced comparable results, although RF modeling showed higher accuracies over thermokarst ponds and open areas. Overall, the paper offers valuable insights into snow cover dynamics over palsas, though minor revisions could enhance its clarity and impact. | Answered | Thank you very much for your constructive comments and the positive overall assessment of our manuscript. We appreciate the suggestions, which have helped to improve the clarity and quality of the paper. Please note that the line numbers referenced in the review do not correspond to the final version of the manuscript as submitted. We have made every effort to match each comment to the relevant section and have indicated the correct line numbers where possible. |
| **Recommendations:** | | |
| 2. Line 6: Change "Digital Surface Model" to "Digital Terrain Model" | Changed | Thank you for pointing this out. We have corrected the term to "Digital Terrain Model" in lines 6 - 7. |
| 3. Lines 15-18: It may be preferable to present the UAS lidar results first and then note the enhancements made by using RF modeling. Differentiating the two could be misleading given that the RF model relies on input parameters derived from UAS lidar data. | Changed | Thank you for this valuable feedback. We have revised the abstract to present the LiDAR results first, followed by the RF modeling results. This reordering better reflects the fact that the RF model relies on input parameters derived from the LiDAR data. |
| 4. Figure 3: Perhaps this figure could have the first row represent the UAS lidar process and the second row represent the RF process? Or maybe specify the input parameters to make it clearer? | Answered/Changed | Thank you for this suggestion. While we appreciate the idea, we believe that emphasizing the data acquisition process rather than the modeling workflow is more appropriate at this point in the manuscript. The UAS LiDAR and RF procedures are |

| | | |
|---|---|---|
| | | described in detail in Section 3. However, to improve clarity, we have added the list of input parameters to the figure to show at which stage they are derived. |
| 5. Figure 5: The negative snow depth from UAS-Lidar is mentioned in the Discussion, but I would suggest briefly making note of it in the Results as well. Also, changing the color of the snow depth points to indicate snow depth measurements could also be useful to directly compare to RF and lidar data. | Changed | Thank you for the helpful suggestion. The requested changes have been implemented in the figure, and a corresponding note on the negative values has been added to the Results section (lines 290–292): "*The UAS LiDAR dataset includes negative snow depth values, which result from elevation mismatches between the summer and winter DTMs. These are visualized in red to distinguish them clearly in Figure 5.*" |
| 6. Table 3: Can you clarify why MAE and SD are missing for each of the 3 study sites? | Answered/Changed | Thank you for your comment. We originally considered MAE and SD to be less informative for the external validation and therefore excluded them. However, to ensure consistency and avoid confusion, we have now added these values to Table 3. |
| 7. Line 464: It could be helpful to quantify this tighter spread, since the numbers in the previous tables seem to indicate that RF would have lower variance. | Changed | Thank you for pointing this out. We acknowledge that we had mistakenly reversed the interpretation of the LiDAR and RF results in the original sentence. This has been corrected in lines 341 - 343: "*The $SD_{RF}$ results exhibit a tighter spread around the regression line, indicating lower variance compared to $SD_{LiDAR}$. This is consistent with the standard deviation values reported in Table 4, where $SD_{RF}$ shows a 13 - 65% lower spread across validation point groups.*" |
| 8. Line 555: Clarify "open water" considering it's still snow covered in the winter dataset. Noting that the lidar is scanning open water in the summer dataset could be helpful to the reader. | Changed | Thank you for the helpful comment. We have replaced "open water" with "thermokarst pond" for greater clarity, and we added the following sentence in lines 387 - 389: *Although these areas are snow-covered during winter LiDAR acquisition, they are characterized by open water surfaces in the summer dataset used to derive snow depth by DTM subtraction.* |
| 9. Line 558: Volumetric scattering does occur when lidar scans a snow surface, but the error is on the scale of <4 cm at this wavelength. I would suggest the difference | Changed | Thank you for this insightful comment. We have addressed this point in lines 393 - 395 with the following sentence: *Even though* |

| | | |
|---|---|---|
| in snow depth (~30 cm) is most likely from scanning the open water during the summer as more absorption is occurring over this surface. | | *volumetric scattering in snow can affect LiDAR results, the associated error at wavelengths commonly used for snow depth measurements, such as the 905 nm wavelength applied in this study, is typically in the low centimeter range (Deems et al., 2013), and thus does not account for the larger discrepancies observed in this study.* |
| 10. Lines 672-673: Perhaps it could be helpful to note the snow conditions at the time of winter data collection, which would also impact scattering and lidar returns (i.e., general age of snow/grain size, presence of light-absorbing impurities, etc.). | Answered/Changed | Thank you for the suggestion. As snowpack properties were not recorded during the field campaign, we cannot provide detailed information on snow conditions. However, we agree that such data could be useful to assess uncertainty in LiDAR-derived snow depth. Therefore, we added the following sentence in lines 473 - 476: *While the wavelength-related interaction with the snow surface is a key factor, detailed information on snow conditions, such as grain size, snow age, or impurity content, was not collected during the field campaign. Such data could, however, help to better assess potential sources of uncertainty in the LiDAR-derived snow depth, particularly those related to scattering and absorption effects.* |
| 11. Lines 678-682: Could you specify which wavelengths would be used to improve snow depth mapping? Visible wavelengths would penetrate deeper into the snowpack, and shortwave infrared has a higher likelihood of absorption (Deems et al., 2013). | Changed | Thank you for the helpful comment. We agree that a clearer specification of relevant wavelengths can improve the understanding of LiDAR snow interactions. We have therefore expanded this section in lines 468 - 473 to recommend the use of shortwave infrared wavelengths, such as 1550 nm, due to their stronger absorption in ice and the resulting surface-confined return signal. This can help reduce uncertainty, particularly over complex or low-reflectivity surfaces. The updated text now reads: *Another source of uncertainty is the choice of LiDAR wavelength. The 905 nm wavelength used in this study is typical for many airborne systems and generally produces only minor depth errors in snow due to limited penetration, with most of the signal returned from the upper centimeters of the snowpack (Deems et al., 2013). In comparison, shortwave infrared wavelengths such as 1550 nm are more* |

| | | |
|---|---|---|
| | | *strongly absorbed by ice, resulting in a return signal that is more confined to the snow surface. This characteristic can help reduce uncertainty, particularly in areas with complex surface conditions or low reflectivity.* |
| **Minor Suggestions:** | | |
| 12. Line 14: "…0.77 and 0.691, respectively." | Changed | Corrected in lines 8 - 12. |
| 13. Lines 130-148: Looks like there's use of both past and present tense, maybe double check for consistency. | Changed | Past tense adjusted for consistency. The sentence "In-situ measured snow depth data was used…" was changed to present tense in line 101. |
| 14. Line 147: "…approach provides the most reliable results, and (2) how do the snow depth patterns…" | Changed | Corrected in line 108. |
| 15. Figure 1: The font in 1b could be larger. The "Road" label could also be replaced with "Route E8" instead. | Changed | Thank you very much for the suggestions. We increased the font size, replaced "Road" with "Route E8", and lake names were removed to improve readability. |
| 16. Line 160: Instead of "in the west" perhaps "extends to the west"? | Changed | We changed it in line 118. |
| 17. Line 249: May want to double-check the grammar on this line ("-,"). | Changed | We changed the sentence in lines 195 – 197: "*Based on the summer and winter DTM of the palsa sites, snow distribution datasets were calculated by substracting the winter by the summer DTM in Geographic Information Systems (GIS) using ArcGIS Pro by Esri, allowing the comparison of UAS LiDAR conducted snow depth SD$_{LiDAR}$ and RF modeled SD$_{RF}$.*". |
| 18. Line 375: I'd recommend using either "Especially" or "directly". | Answered | Unfortunately, the line number indicated does not match the corresponding section in the manuscript, and despite careful review, we were unable to identify the specific sentence referred to in this comment. |
| 19. Figure 6: Perhaps it could be useful to clip the results to the palsa boundaries (or at least a palsa buffer) rather than the raster extent. | Answered | We acknowledge the suggestion to clip the results to the palsa boundaries or a surrounding buffer. However, we decided to retain the full raster extent in Figure 6, as it provides a more comprehensive visual comparison between the RF and LiDAR approaches. Clipping the results would exclude relevant areas at the palsa edges, which are important for interpreting snow accumulation patterns and include several |

| | | measurement points. Additionally, the figure illustrates that the LiDAR data covers areas not included in the RF training data, thereby highlighting differences in spatial coverage and model performance. For these reasons, we decided to keep the original extent. |
|---|---|---|
| 20. Line 490: "indicate" rather than "indicating"; RF may improve the lidar results, but I would be hesitant to present the results as distinct. | Changed | The verb was changed from "indicating" to "indicate" in lines 341–345. |
| 21. Figure 9: Maybe "demonstration" rather than "explanation"? | Changed | We changed it to demonstration. |
| 22. Line 732: Similar to the previous comments, perhaps "alternative" could be rephrased as this suggests that it's an independent method. | Changed | "Alternative" was replaced with "approach" in line 505 to avoid implying that the method is fully independent. |
| 23. Figure A2: I would suggest differentiating the colors more, especially the white and yellow points. | Changed | The color of the "Thermokarst" class was changed from yellow to orange in Figure A2 to improve visual differentiation. |

**Appendix**

This section provides revised figures in response to comments #4, #5, #15, and #23.

**Appendix A**

[Figure]

**Appendix B**

[Figure]

**Appendix C**

[Figure]

**Appendix D**

[Figure]

Point Class
- Edge
- On Top
- Open Area
- Thermokarst

☐ Palsa edge in 2022

**Background**
UAS orthomosaic
25.08.2022